# Persistent Homology for Time Series: a Selective Review

**Alexandre Bois**                                                    *alexandre.bois@ens-paris-saclay.fr*
*Université Paris-Saclay, Université Paris Cité, ENS Paris-Saclay, CNRS, SSA, INSERM, Centre Borelli*

**Hugo Henneuse**                                             *hugo.henneuse@universite-paris-sacalay.fr*
*Université Paris-Saclay, Université Paris Cité, ENS Paris-Saclay, CNRS, SSA, INSERM, Centre Borelli*

**Brian Tervil**                                                        *rian.tervil@ens-paris-saclay.fr*
*Université Paris-Saclay Université Paris Cité, ENS Paris-Saclay CNRS, SSA, INSERM, Centre Borelli*

**Laurent Oudre**                                                    *laurent.oudre@ens-paris-saclay.fr*
*Université Paris-Saclay Université Paris Cité, ENS Paris-Saclay CNRS, SSA, INSERM, Centre Borelli*

**Reviewed on OpenReview:** *https://openreview.net/forum?id=tztKO9jzBR&noteId=Kw6Vw8Hm1S*

## Abstract

Over the last ten years, persistent homology has been increasingly used to analyze the structure and shape of various types of data, including time series. This article is a review of persistent homology applied to (univariate or multivariate) time series data. We review 87 articles that apply methods involving persistent homology to time series data, published between 2014 and 2025 in several domains of application, such as biomedicine, industry, and economics. We introduce the main concepts of persistent homology, give an overview of the application fields and tasks, and propose a general framework to describe the main characteristics of all the methods.

## 1 Introduction

Topological data analysis (TDA) is a research area that uses tools from algebraic topology to study the shape of data. More precisely, TDA assumes the existence of an underlying set (manifold, continuous function) from which our data are a finite sample, and seeks to recover information about its topology by structuring the data. Persistent homology (Edelsbrunner & Harer, 2010; Boissonnat et al., 2018) is a popular tool of TDA that has had many applications (Chazal & Michel, 2021; Skaf & Laubenbacher, 2022) in the last decades. The idea of persistent homology is to build a multi-scale structure on data by constructing a sequence of simplicial complexes (a filtration) and to study features (homological classes) that persist through many scales. Theoretical results, such as stability theorems (Chazal et al., 2014), guarantee that persistent homology is relevant to handle noisy data.

Recently, an increasing number of studies have used persistent homology to analyze time series in several domains of application, such as biomedicine, industry, and economics. Existing reviews on persistent homology for time series (Seversky et al., 2016; Gholizadeh & Zadrozny, 2018; Ravishanker & Chen, 2019; Perea, 2019; Xu et al., 2021; Hernández-Lemus et al., 2024) either focus on one application (cardiology, neuroscience), do not mention certain important articles and methods, or do not describe a general framework to identify different types of methods. Table 1 sums up the characteristics of existing reviews and ours. The "Type of methods" column indicates if a review deals with univariate time series only or univariate and multivariate, and if the reviewed methods focus on a specific type of data transformation (delay embedding, graph, or other). The "Framework" column indicates if the review defines a general framework to decompose methods into several steps. The "Number of references" column refers to the number of cited articles that directly apply TDA to time series: articles introducing the fundamentals of TDA or applying other methods to time series are not counted.

| Review | Year | Data | Methods | Framework | #Refs |
|---|---|---|---|---|---|
| Seversky et al. (2016) | 2016 | General | Delay embedding (uni- & multivariate) | Yes | 8 |
| Gholizadeh & Zadrozny (2018) | 2018 | General | Delay embedding + graphs | No | 22 |
| Ravishanker & Chen (2019) | 2019 | General | General (univariate) | Yes | 12 |
| Perea (2019) | 2019 | General | Delay embedding (uni- & multivariate) | No | 15 |
| Xu et al. (2021) | 2021 | EEG | General | Yes | 12 |
| Hernández-Lemus et al. (2024) | 2024 | Cardiovascular | General | Yes | 12 |
| **Ours** | 2024 | General | General | Yes | **87** |

Table 1: Comparison of existing literature reviews.

This article reviews studies applying persistent homology to the analysis of time series. While not exhaustive, our aim is to summarize the main trends in the field and provide readers with a clear and comprehensive overview of the key ideas and methods developed to date. The selection process began with papers identified through a Google Scholar search using the query "persistent homology time series". It was then significantly expanded by incorporating additional works that either cite these initial papers or are cited by them, thereby capturing a broader and more representative subset of the literature. Note that topological methods other than persistent homology are also labeled as TDA, such as Mapper (Singh et al., 2007), UMAP (McInnes et al., 2018), simplicial complex signal processing (Barbarossa & Sardellitti, 2020), or sheaves-based approaches (Robinson, 2014). The scope of this review is limited to persistent homology.

The paper is organized as follows. Section 2 gives an overview of the application fields of persistent homology for time series. Section 3 introduces the main mathematical notions evoked in the rest of the paper. In Section 4, we propose a general framework to describe the main characteristics of the approaches in the 87 selected articles, which we organize thematically. Finally, in Section 5, we discuss some gaps in the literature and outline perspectives for future research.

## 2 Application fields and tasks

Persistent homology has been applied to time series from many fields. Within the 87 selected papers, the vast majority have biomedical applications. In particular, cardiology and neurology are by far the most represented applications, and neurophysiology (the study of neurological diseases through physiological time series such as motion sensor data). This can be explained by the fact that biomedical signals are often very structured (periodicity for ECGs and motion signals, correlations for EEGs), which is suitable for TDA-based methods (more concrete details will be provided in section 4), and by the fact that biomedicine is an important field of research in data analysis in general. For example, persistent homology was used to improve ECG classification for arrythmia detection (Umeda, 2017; Liu et al., 2023) or atrial fibrillation detection and classification (Safarbali & Golpayegani, 2019; Jiang et al., 2022), and to define visualization methods based on fMRI data that enabled researcher to better understand changes in brain connectivity during motor learning (Stolz et al., 2017) or after consuming psychedelic mushrooms (Petri et al., 2014).

Another important field is dynamical systems, often with industrial applications such as chatter detection in turning processes (Khasawneh & Munch, 2016; Khasawneh et al., 2018; Yesilli et al., 2022). This is mainly because data from this field are also very structured, with different regimes such as regular behavior and chaotic behavior. As explained in Section 4.3, delay embeddings are a popular tool to study dynamical systems, and persistent homology can be useful to study their topology.

Persistent homology has also been applied to detect periodicity in video and motion capture data, classify or identify musical signals (Sanderson et al., 2017; Reise et al., 2024), detect events or change in financial time series (Ueda et al., 2022; Gidea, 2017; Gidea & Katz, 2018; Gidea et al., 2020), or classify/cluster time

series from biology (Corcoran & Jones, 2017), ecology (Chen & Ravishanker, 2023) and geography (Chen et al., 2019). Persistent homology is also a subject of mathematical study (Perea & Harer, 2015; Perea, 2016; Gakhar & Perea, 2024; Chazal et al., 2025).

The field of each article can be found in Tables 2, 3, 4.

Persistent homology was used to create new algorithms for generic tasks in time series analysis, such as change point detection (Ueda et al., 2022; Fernández et al., 2023), anomaly detection (Bois et al., 2024a; Chazal et al., 2024), motif detection (Germain et al., 2024), time series forecasting (Zeng et al., 2021), periodicity quantification (Dłotko et al., 2019) or zero-crossing detection (Tanweer et al., 2024). In concrete applications, other common machine learning or data analysis tasks are performed on features from persistent homology, such as time series classification using machine learning classifiers or deep learning, statistical analysis, or data visualization. Note that most methods are generic and can be applied to other fields than the one they were introduced in.

## 3 Background on persistent homology

Here, we introduce mathematical objects from persistent homology. We refer to Edelsbrunner & Harer (2010); Boissonnat et al. (2018) for a more complete introduction and to Chazal et al. (2014; 2016) for more theoretical aspects.

### 3.1 Simplicial complexes and filtrations

We first introduce the notion of **simplicial complex**, a mathematical structure used to approximate the shape of a dataset by connecting points with segments, triangles, tetrahedra, and higher-dimensional analogues. These simple building blocks (called simplices) are assembled according to specific rules (see the following definition) to capture topological features of the data. Importantly, their combinatorial and discrete nature makes them well suited for the computation of topological invariants. See Figure 2 for an illustration of a simplicial complex.

**Definition 1** (Simplicial complexes). *A $k$-simplex on a set $X \subset \mathbb{R}^d$ is an unordered tuple $\sigma = [x_0, ..., x_k]$ of $k+1$ distinct elements of $X$. The elements $x_0, ..., x_k$ are called the vertices of $\sigma$. If each vertex of a simplex $\rho$ is also a vertex of $\sigma$, then $\rho$ is called a face of $\sigma$. A simplicial complex $K$ is a set of simplices such that any face of a simplex of $K$ is a simplex of $K$.*

The main idea of persistent homology is to move beyond the analysis of a single fixed topological space and instead capture the evolution of topological features across a nested family of spaces. Such a family is typically represented by an increasing sequence of simplicial complexes, called a **filtration**, which encodes the data at multiple scales.

**Definition 2** (Filtration). *A filtration of a simplicial complex $K$ (or filtered simplicial complex) is a family of simplicial complexes $(K_\alpha)_{\alpha \geq 0}$ such that $K_0 = \emptyset$, $\alpha < \alpha' \Rightarrow K_\alpha \subset K_{\alpha'}$ and $\bigcup_{\alpha \geq 0} K_\alpha = K$.*

The **filtration value** of a simplex $\sigma \in K$ is the lowest $\alpha$ such that $\sigma \in K_\alpha$. When the number of simplices is finite, we will only use a finite set of indices $\alpha_i$ such that $\emptyset = K^{\alpha_0} \subset K^{\alpha_1} \subset \cdots \subset K^{\alpha_m} = K$ and $\alpha_i \leq \alpha < \alpha_{i+1} \Rightarrow K_{\alpha_i} = K_\alpha$. Without loss of generality, we can also assume that for all $i$ there exists a simplex $\sigma_{i+1} \in K$ such that $K^{\alpha_{i+1}} = K^{\alpha_i} \cup \{\sigma_{i+1}\}$. In the remainder of this section, we discuss several key filtrations applicable to different types of data structures.

**Sublevel set filtration.** The most direct way to apply persistent homology to time series, as further described in Section 4.2, is to filter the series (or a portion of it) using its sublevel sets. To formalize this idea, let $f : X \to \mathbb{R}$ be a real-valued function defined on some set $X \subset \mathbb{R}^d$. Inspired by Morse theory (Milnor, 1963), a natural way of constructing a filtration on $X$ is to consider the sublevel sets $f^{-1}(]-\infty, \alpha])$ for increasing values of $\alpha \in \mathbb{R}$. See Figure 1 for an illustration.

**Definition 3.** *The $d$-dimensional simplicial complex $K_\alpha$ is then defined by:*

$$\sigma \in K_\alpha \Longleftrightarrow \forall v \in \sigma, \ v \in f^{-1}(]-\infty, \alpha]).$$

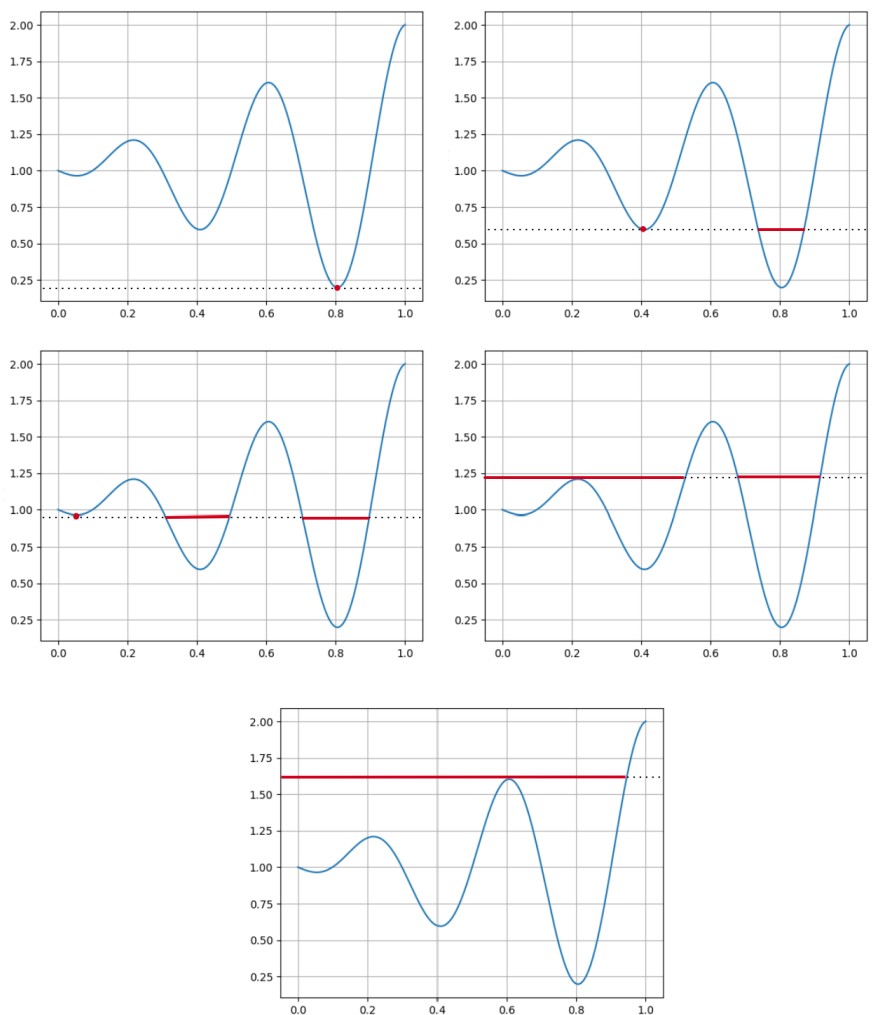

Figure 1: Sublevel set filtration of a simple time series. The sublevel sets are shown in red at critical levels ($\lambda_1 \approx 0.2$, $\lambda_2 \approx 0.6$, $\lambda_3 \approx 0.9$, $\lambda_4 \approx 1.2$ and $\lambda_5 \approx 1.6$) where either a new connected component appears or two connected components merge.

*The collection $(K_\alpha)_{\alpha \in \mathbb{R}}$ is called the sublevel sets filtration associated to $f$.*

Note that if instead of the sublevel sets we are interested in the topology of the superlevel sets, we can define a filtration similarly by simply replacing $f^{-1}(]-\infty, \alpha])$ by $f^{-1}(]\alpha, +\infty])$.

**Čech and Rips filtrations.** For point cloud data, as is commonly obtained from a time series through delay embedding (see Section 4.3), two standard filtrations are the Čech and Vietoris-Rips (or Rips, VR) filtrations. In what follows, we let $X = \{x_1, \dots, x_n\}$ be a point cloud and $B_\alpha(x)$ be the open ball of radius $\alpha$ and centered at $x$.

**Definition 4.** *The Čech complex $\mathsf{Cech}(X, \alpha)$ is defined by:*

$$[x_{i_1}, \dots, x_{i_k}] \in \mathsf{Cech}(X, \alpha) \iff \bigcap_{j=1}^{k} B_\alpha(x_{i_j}) \neq \emptyset.$$

The nerve theorem, a result in algebraic topology, implies that if the balls $B_\alpha(x)$ have empty or contractible (homotopy equivalent to a point) intersections, then $\mathsf{Cech}(X, \alpha)$ is homotopy equivalent to $\bigcup_x B_\alpha(x)$. This means that the simplicial complex carries the topology of the cover of X by open balls. The nerve theorem

is also used to prove the theorem from Niyogi et al. (2008) which, as mentioned above, guarantees that there exists a scale such that $\mathsf{Cech}(X, \alpha)$ is homotopy equivalent to the topological space of which $X$ is a sampling.

**Definition 5.** *The Rips complex* $\mathsf{Rips}(X, \alpha)$ *is defined by:*

$$[x_{i_1}, \ldots, x_{i_k}] \in \mathsf{Rips}(X, \alpha) \Longleftrightarrow \forall j, l, \ \|x_{i_j} - x_{i_l}\| \leq \alpha$$

Figure 2 shows a cover of a point cloud by a union of balls with the same radius, and the corresponding Čech and Rips complexes. Notice that the three lower balls have an empty intersection, but each pair of balls intersects, so the triangle in the Rips complex is not present in the Čech complex.

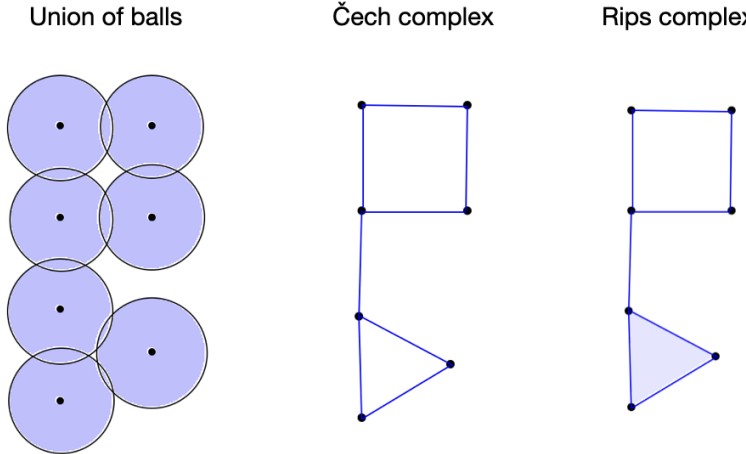

Figure 2: An example of a cover of a point cloud by a union of balls, and the corresponding Čech and Rips complexes.

The Rips filtration and the Čech filtration are obtained as explained above from the nested families of simplicial complexes. The Rips filtration is easier to compute than the Čech filtration because only distances are needed instead of intersections of balls.

**Clique filtration.** Our final example deals with network data, represented by a (weighted) graph, as often encountered in analyses of the covariance structure of multivariate time series (see Section 4.4). In this context, an important filtration is the clique filtration. Let $G = (V, E, w)$ be a weighted graph, where $V$ is the set of vertices, $E \subset V \times V$ is the set of edges, and $w : E \to \mathbb{R}_+$ is a weight function.

**Definition 6.** *The ascending clique filtration is defined as follows: all the 0-simplices have a filtration value of 0 and, for $\alpha > 0$ and $k \geq 1$:*

$$[v_0, \ldots, v_k] \in K_\alpha \Longleftrightarrow \forall i \neq j, \ (v_i, v_j) \in E \ and \ w(v_i, v_j) \leq \alpha.$$

*The descending clique filtration is defined the same way, by adding all the vertices at the beginning and working with decreasing values of $\alpha$.*

Note that the ascending clique filtration is not exactly a sublevel set filtration because the function is defined on edges instead of vertices. Also note that if $V$ is a metric space and $w$ is its distance, then the clique filtration can be seen as a Rips filtration.

Of course, this list is not exhaustive, and many other filtrations can be considered, as we will see throughout this survey. For additional examples of filtrations, we refer the reader to Boissonnat et al. (2018); De Silva & Carlsson (2004); Anai et al. (2020).

### 3.2 Simplicial homology

Simplicial homology is a fundamental concept that provides a framework for describing the topology of a simplicial complex through algebraic structures. At a high level, it associates algebraic objects (groups or vector spaces) to a simplicial complex in order to quantify features like connected components, holes, and higher-dimensional voids. For instance, the 0-th homology group represents the connected components of the complex, the 1-th homology group represents its loops and the 2-th homology group represents its cavities. The rest of this section introduces the minimal formalism needed to rigorously define this concept.

Let $K$ be a simplicial complex with maximal simplex dimension $d$, $\mathbb{G}$ be an abelian group, and $0 \le k \le d$.

**Definition 7.** *The space $C_k(K; \mathbb{G})$ of k-chains is defined as the set of formal sums of k-simplices of $K$ with coefficients in $\mathbb{G}$, that is to say, if all the k-simplices of $K$ are $\sigma_1, \ldots, \sigma_{n_k}$, all the elements of the form:*

$$c = \sum_{i=1}^{n_k} a_i \sigma_i, \quad a_i \in \mathbb{G}.$$

$C_k(K)$ is an abelian group whose addition is naturally defined: if $c = \sum_{i=1}^{n_k} a_i \sigma_i$ and $c' = \sum_{i=1}^{n_k} a'_i \sigma_i$ then:

$$c + c' = \sum_{i=1}^{n_k} (a_i + a'_i) \sigma_i.$$

Using a group $\mathbb{G}$ gives the more general definition, but from now on, we will only consider $\mathbb{G} = \mathbb{Z}/2\mathbb{Z}$, so the coefficients are modulo 2, which allows us to avoid orientation considerations. $C_k(K)$ is now a vector space, with scalar multiplication defined as $\lambda c = \sum_{i=1}^{n_k} (\lambda a_i) \sigma_i$ for $\lambda \in \mathbb{Z}/2\mathbb{Z}$. Furthermore, choosing $\mathbb{G} = \mathbb{Z}/2\mathbb{Z}$ admits a natural geometric interpretation of chains. More precisely, a $k$-chain can be identified with a finite collection of $k$-simplices, and the sum of two $k$-chains corresponds to the symmetric difference of the two associated collections.

**Definition 8.** *Let $\sigma = [v_1, \ldots, v_k]$ be a k-simplex with vertices $v_1, \ldots, v_k$, and $[v_1, \ldots, \hat{v}_i, \ldots, v_k]$ be the $(k-1)$-simplex spanned by those points minus $v_i$. The boundary operator $\partial$ is defined as:*

$$\partial : \begin{cases} C_k(K) & \longrightarrow & C_{k-1}(K) \\ \sigma & \longmapsto & \partial \sigma = \sum_{i=1}^{k} (-1)^i [v_1, \ldots, \hat{v}_i, \ldots, v_k]. \end{cases}$$

We have the following sequence of linear maps:

$$C_d(K) \xrightarrow{\partial} C_{d-1}(K) \xrightarrow{\partial} \ldots \xrightarrow{\partial} C_1(K) \xrightarrow{\partial} C_0(K) \xrightarrow{\partial} \{0\}.$$

They satisfy $\partial \circ \partial = 0$ : we call such a sequence of maps a chain complex. This constitutes the setup for homology. We can now define cycles and boundaries, homology groups, and Betti numbers.

**Definition 9.** *We define the set $Z_k(K)$ of k-cycles of $K$ as :*

$$Z_k(K) = \mathsf{Ker}(\partial : C_k(K) \to C_{k-1}(K))$$

*and the set $B_k(K)$ of k-boundaries of $K$ as :*

$$B_k(K) = \mathsf{Im}(\partial : C_{k+1}(K) \to C_k(K)).$$

*We have $B_k(K) \subset Z_k(K) \subset C_k(K)$ (Boissonnat et al., 2018) so we can define the $k^{th}$ homology group as:*

$$H_k(K) = Z_k(K)/B_k(K)$$

*and the $k^{th}$ Betti number:*

$$\beta_k(K) = \mathsf{dim}(H_k(K)).$$

A $k$-cycle is a "loop" made of $k$-simplices. A $k$-boundary is a $k$-cycle made of all the faces of $k + 1$-simplices in $K$. A $k$-dimensional "hole" is a cycle that is not a boundary. Each hole is represented by a homology class in $H_k(K)$, $\beta_k$ represents the number of $k$-dimensional "holes". For example, $\beta_0$ is the number of connected components of $K$, $\beta_1$ is the number of loops, and $\beta_2$ is the number of voids.

On the examples from Figure 2, the four highest edges and the three lowest ones form 1-cycles of the Čech and Rips complex. In the Rips complex, the low one is the boundary of a triangle, which is not the case for the Čech complex. Thus, the Betti numbers of the Čech complex are $\beta_0 = 1$ and $\beta_1 = 2$, and for the Rips complex $\beta_0 = 1$ and $\beta_1 = 1$.

### 3.3 Persistence diagrams and barcodes

Persistent homology studies how the homology groups of a space evolve as the filtration parameter increases. It captures the appearance and disappearance of topological features across scales, producing a discrete summary known as a persistence diagram (or equivalently, barcodes). In this summary, each feature is represented by its **birth**, i.e the point at which it appears (e.g., a new connected component emerges or a loop forms) and its **death**, i.e. the point at which it disappears (e.g., two connected components merge or a loop is filled). This provides a compact, scale-invariant representation of the topological structure of the data. The canonical forms of filtered complexes introduced in Barannikov (1994) encode pairings of the appearance and disappearance of cycles and correspond to what are now known as persistence barcodes and diagrams.

More formally, let $(K^{\alpha_i})_{0 \leq i \leq m}$ be a filtration such that for each index $i$, we go from $K^{\alpha_i}$ to $K^{\alpha_{i+1}}$ by adding a simplex $\sigma_{i+1}$ to $K^{\alpha_i}$. We call $C_k^i, Z_k^i, B_k^i, H_k^i, \beta_k^i$ the respective spaces of $k$-chains, $k$-cycles, $k$-boundaries, $k^{th}$ homology group and $k^{th}$ Betti number of $K^{\alpha^i}$. The goal is to follow the evolution of $H_k^i$ as $i$ increases. It can be shown (Boissonnat et al., 2018) that when a $k$-simplex $\sigma_{i+1}$ ($k > 0$) is added, it either creates a new homology class in $H_k^{i+1}$ (i.e. a new $k$-cycle that is independent of those of $H_k^i$) or it closes a $k-1$-dimensional hole of $H_{k-1}^{i-1}$, so $H_{k-1}^i$ has one less homology class than $H_{k-1}^{i-1}$, in that case we say that $\sigma_{i+1}$ killed a homology class (by convention, we always consider that when two classes merge, the younger class gets killed). If $k = 0$, each new vertex creates a homology class in $H_0$.

The final result of persistent homology is the set of all so-called **persistent pairs** $(\sigma_{l(j)}, \sigma_j)$ such that for each $j$, $\sigma_{l(j)}$ creates a topological fetaure and $\sigma_j$ kills it. We say that the birth time of the associated topological features is $l(j)$, it death time is $j$ and its persistence (or lifetime) is $j - l(j)$. Although formulated in a different framework, the computation of persistence pairs was already considered in the pioneering work of Barannikov (1994). For a modern treatment, we refer the reader to Boissonnat et al. (2018), which provides a detailed presentation of the algorithmic foundations and practical procedures for computing persistence pairs. The k-dimensional **persistence diagram (PD)** is the set of points of coordinates $(\alpha_{l(j)}, \alpha_j)$ such that $\sigma_{l(j)}$ is a $k$-simplex (counted with multiplicity). The points of the diagonal $y = x$ are added with infinite multiplicity (it is useful to define distances). The **persistence barcode** is a representation equivalent to the persistence diagram, where each pair $(\sigma_{l(j)}, \sigma_j)$ is represented as a line that starts at $l(j)$ and ends at $(j)$.

Figures 3a and 3b show the persistence diagram and the barcode associated with the superlevel set filtration depicted in Figure 1. For $\lambda < \lambda_1 \approx 0.2$, the superlevel sets are empty and therefore contain no topological features. At $\lambda_1$, a first connected component appears, giving rise to the point $(\lambda_1, +\infty)$ in the persistence diagram (since this component cannot merge with any older one, its death time is $+\infty$). At $\lambda_2 \approx 0.6$ and $\lambda_3 \approx 0.9$, two additional connected components appear. These components subsequently merge at $\lambda_4 \approx 1.2$. Adopting the convention that the most recently born component dies at a merging event, this produces the point $(\lambda_3, \lambda_4)$ in the persistence diagram. Finally, at $\lambda_5 \approx 1.6$, all remaining connected components merge; all but the oldest component die, yielding the point $(\lambda_2, \lambda_5)$ in the persistence diagram.

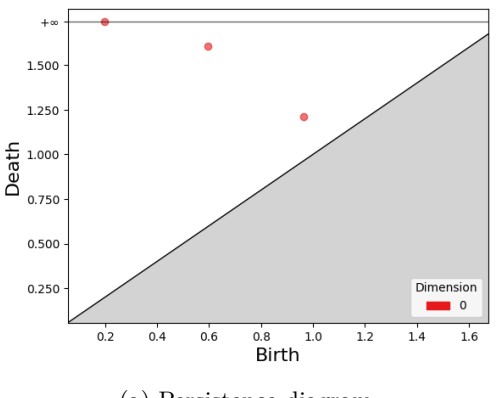

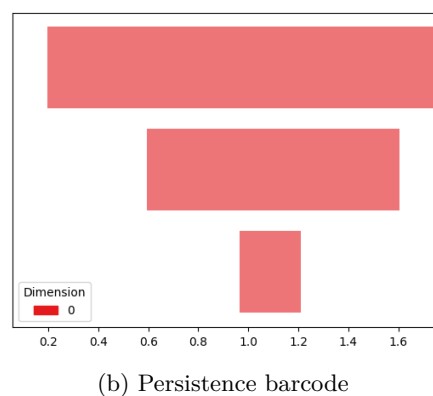

(a) Persistence diagram

(b) Persistence barcode

Figure 3: Persistence diagram and barcode (0D) from the filtration of Figure 1.

### 3.4 Distance between persistence diagrams.

Comparing persistence diagrams (or equivalently, barcodes) requires introducing a suitable notion of distance. Among the various possibilities, the bottleneck distance is particularly popular in statistical contexts.

**Definition 10.** *Let $D$ and $D'$ be two persistence diagrams. The* bottleneck distance *between $D$ and $D'$ is defined as*

$$d_B(D, D') = \inf_{\gamma \in \Gamma(D, D')} \sup_{p \in D} \|p - \gamma(p)\|_\infty,$$

*where $\Gamma(D, D')$ denotes the set of all bijections between $D \cup \Delta$ and $D' \cup \Delta$, and*

$$\|p - q\|_\infty = \max(|x_p - x_q|, |y_p - y_q|),$$

*for points $p = (x_p, y_p)$ and $q = (x_q, y_q)$. Here, $\Delta = \{(x, x) \in \mathbb{R}^2\}$ denotes the diagonal.*

Informally, the bottleneck distance can be seen as a minimal-cost matching distance between the points of two diagrams, with the subtlety that unmatched points are paired with the diagonal $\Delta$. Its popularity is largely due to its stability properties, which make the bottleneck distance robust to small perturbations in the underlying filtration. The foundations of these stability properties can be traced back to Barannikov (1994). The result below offers a contemporary statement of the stability theorem for persistence diagrams associated with filtrations induced by sublevel sets of real-valued functions. More general stability results, encompassing broader classes of persistence modules, can be found in (Chazal et al., 2009; 2016).

**Theorem 1.** *(Cohen-Steiner et al., 2007) Let $\mathcal{X} \subset \mathbb{R}^d$ a compact sets and $f$ and $g$ two continuous functions from $\mathcal{X}$ into $\mathbb{R}$. We have:*

$$d_B(D_f, D_g) \leq \|f - g\|_\infty,$$

*with $D_f$ (resp. $D_g$) denotes the persistence diagram associated with the sub-level sets filtration of $f$ (resp. $g$).*

In addition to the $L_\infty$ matching underlying the bottleneck distance, other norms on $\mathbb{R}^2$ can be used to define, in a similar manner, Wasserstein distances between persistence diagrams.

**Definition 11.** *For a given norm $\|.\|_l$, the $q$-Wasserstein distance is defined as:*

$$d_{W_{q,l}}(D, D') = \inf_{\gamma \in \Gamma(D, D')} \left( \sum_{p \in D} \|p - \gamma(p)\|_l^q \right)^{\frac{1}{q}}.$$

Although general Wasserstein distances do not enjoy stability properties as strong as those of the bottleneck distance, they still possess useful stability guarantees (Skraba & Turner, 2020). An advantage of Wasserstein

distances is that they downweight less the small topological features near the diagonal, which are often regarded as noise but can also contain valuable information about the geometry of the underlying objects (Bubenik et al., 2020b).

### 3.5 Alternative representations

Several alternative representations can be derived from a persistence diagram. For instance, a popular representation called **persistence landscapes (PL)** (Bubenik, 2015) can be obtained by transforming the diagram into a set of piecewise-affine functions, each of which is referred to as a landscape.

**Definition 12.** *For each point $(b, d)$ in a persistence diagram $D$, we define the function $f_{b,d} : \mathbb{R} \longrightarrow [0, \infty]$ as:*

$$
f_{b,d}(x) = \begin{cases} 0 & if \quad x \notin [b, d] \\ x - b & if \quad x \in [b, \frac{b+d}{2}] \\ d - x & if \quad x \in [\frac{b+d}{2}, d] \end{cases}
$$

*The $k$-th persistence landscape of $D$ is the function:*

$$
\lambda_k : \begin{cases} \mathbb{R} & \longrightarrow & [0, \infty] \\ x - b & \longmapsto & \mathsf{kmax}(\{f_{b,d}(x)\}_{(b,d) \in D}) \end{cases}
$$

*Where $\mathsf{kmax}(S)$ is the $k$-th largest value of the set $S$.*

Landscapes represent persistence in a vector space of functions, so $L^p$ norms and the induced distances can be defined to compare landscapes. From a statistical perspective, this "vectorization" of the persistence diagram is appealing, as it allows for the straightforward computation of standard statistics; for example, it is possible to average landscapes. Figure 4 shows an example of a persistence landscape.

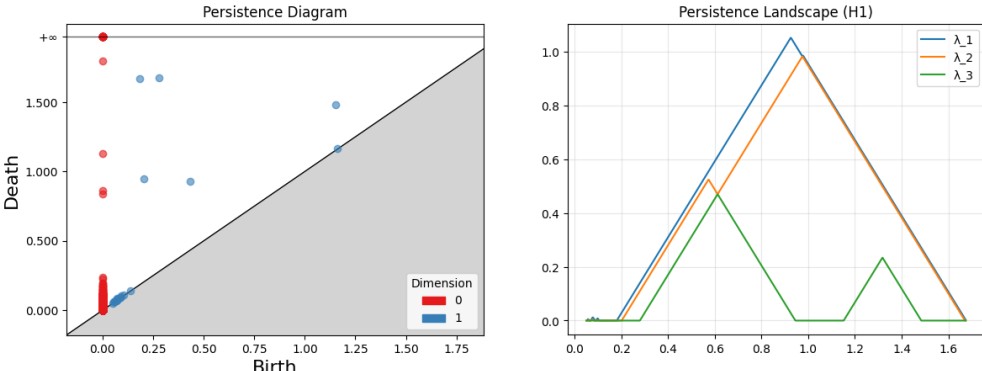

Figure 4: A persistence diagram and the first three associated 1D persistence landscapes.

Another prominent vectorization, especially in the context of time series analysis, is the Betti Curves, introduced by Umeda (2017).

**Definition 13.** *The $k$-th Betti curve of a filtered simplicial complex is the function $\alpha \mapsto \beta_k(K_\alpha)$.*

Beyond those important examples, several other vectorizations have been defined, such as persistence images, silhouettes, paths, and kernels. These representations are more suitable than persistence diagrams for analysis with usual machine learning methods. See Ali et al. (2023) for a survey on vectorization methods where the above vectorizations are defined, and Hensel et al. (2021) for a survey on TDA for machine learning.

## 4 Methods

This section is a review of the articles using persistent homology for time series analysis. We first introduce the typical pipeline that is common to all methods, then review methods for univariate and multivariate

data. Two subsections are dedicated to univariate time series, corresponding to the two main categories of methods we distinguished: those using sublevel set filtrations and 0D persistent homology, and those that transform time series into a point cloud via a delay embedding before studying persistent homology in higher dimensions. A small number of methods do not exactly fit in those two categories; we describe them along with the most related methods. The last subsection is about multivariate time series.

## 4.1 General framework

Throughout this review, we refer to a univariate time series (or signal) as a time-indexed sequence of real numbers $x = (x[t])_{t \in I}$, where $I$ is a finite index set. A multivariate time series with $m$ channels is defined as a collection of $m$ univariate time series sharing the same index set $I$, and is written as $\mathbf{x} = (\mathbf{x}[t])_{t \in I} = ((x_1[t], \ldots, x_m[t]))_{t \in I}$. For clarity, we restrict the discussion to uniformly sampled time series, where the time interval between two consecutive observations is constant, so that the index set can be written $I = [1, n]$, with $n$ denoting the series length. This assumption is made solely for notational convenience and to provide a common framework across the 87 papers considered. In practice, several presented approaches can be applied, or readily adapted, to non-uniformly sampled data.

Each method of our review can be decomposed using the following framework:

$$\text{time series} \to \text{transformation} \to \text{filtration} \to \text{TDA tools} \to \text{analysis}$$

see Figures 5, 7 and 8 for illustrations. The input data is always a time series (univariate or multivariate). It can be transformed into another object on which a filtration is defined. Then, persistent homology can be studied with various TDA tools (persistence diagrams, vectorizations, homology representatives...), and then a specific analysis (statistics, visualization, machine learning...) is performed. Tables 2, 3, 4 list all the 87 methods reviewed here and decompose them in our framework. We also specify when articles include theoretical contributions, such as new tools or mathematical results.

## 4.2 Sublevel sets methods

We first review methods involving the sublevel set filtration, which provides the most direct way to apply persistent homology to time series. Figure 5 illustrates a typical framework using sublevel sets. This filtration has been particularly popular in the analysis of univariate time series $x = (x_i)_{1 \le i \le n}$ (seen as a real-valued function on $\mathbb{N}$). In this context, $[i]$ is a vertex of $K_\alpha$ if and only if $x_i \le \alpha$ and $[i, j]$ is an edge of $K_\alpha$ if and only if $\max(x_i, x_j) \le \alpha$. This filtration can be computed in $\mathcal{O}(n)$ and its diagram in $O(n \log(n))$. Furthermore, as highlighted in the background section, the obtained diagram enjoyed nice stability properties. Interestingly, the persistence diagram consists of pairs $(i, j)$ such that $x_i$ is a local minimum and $x_j$ is a local maximum. Alternatively, if one considers the superlevel set filtration instead of the sublevel set filtration, the diagram consists of pairs $(i, j)$ such that $x_j$ is a local minimum and $x_i$ is a local maximum. Although this transformation may discard certain information about the signal, the resulting persistence diagram still captures essential structural features and remains highly interpretable for a wide range of practical tasks. We now highlight some important themes related to this approach.

**Topological filtering, extrema detection, and application to truncated time series.** Following the previous remark, a natural application of persistent homology to time series is the detection of local minima and maxima. Beyond stability, this requires the ability to distinguish features arising from noise from true topological features in empirical persistence diagrams. In general statistical contexts, this question has received considerable attention, leading to the development of several "topological filtering" approaches and thorough statistical studies (Chazal et al., 2013; Fasy et al., 2014; Bubenik, 2015). The simplest approach consists of removing points in the diagrams that lie within a certain distance of the diagonal. In the more specific context of time series analysis, Myers et al. (2020) proposes such an approach for different additive noise models and provides statistical guarantees. They highlight that this consequently yields a robust method for detecting local extrema.

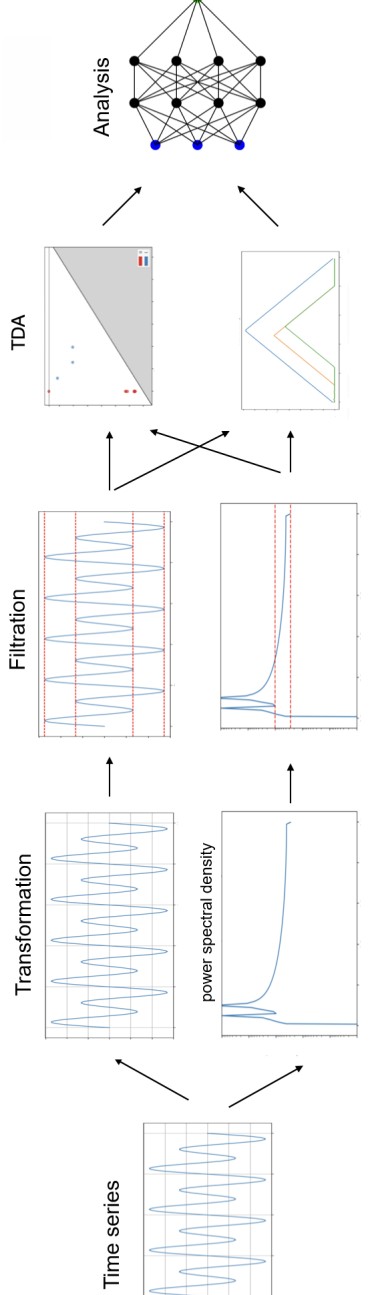

Figure 5: Typical framework of persistence homology for time series using sublevel sets.

Similarly, sublevel set filtrations have also been used on truncated time series (time series where points under or above a certain value are ignored) to create robust algorithms. One of the first such applications was proposed by Khasawneh & Munch (2018), who introduced an algorithm for counting pulses in piecewise constant (binary) industrial signals affected by noise, "digital ringing" (spurious pulses occurring near genuine ones), and variable inter-pulse intervals. They define a model of their signals and use it to derive a theorem stating that with small noise, dense enough sampling, and enough visible pulses, their algorithm counts the right number of pulses. More recently, in Tanweer et al. (2024), sublevel set filtrations are studied separately for the positive and negative values of the time series to find zero-crossings in noisy time series. They threshold both persistence diagrams and merge the remaining values to define intervals with potential zero crossings. Thresholding the diagrams makes the method robust by avoiding finding multiple crossings around each true one due to noisy oscillations.

**Analysis of periodic time series.** The persistence diagrams obtained from sublevel set filtrations are also invariant under reparametrization: the persistence diagrams are the same for a sampling of $f : \mathbb{R} \to \mathbb{R}$ and of $f \circ \gamma$, where $\gamma : \mathbb{R} \to \mathbb{R}$ is an increasing bijection. This property was used in Bonis et al. (2024) and later in Chazal et al. (2025) to define methods to study noisy reparameterized samplings of periodic functions. These articles also highlight the additivity property of the sublevel set filtration of periodic functions: the persistence diagram corresponding to the sublevel filtration of $N$ periods is related to $N$ times the persistence diagram corresponding to one period. In Bonis et al. (2024), the authors introduce an algorithm to count the number of periods and find an odometric sequence for reparametrised periodic signals, for an industrial application (magnetic odometry). They use the fact that, seeing a periodic function as a function on the circle, the diagram of sublevel sets of $N$ periods is equal to $N$ times the diagram of one period, so they count the multiplicity of points on the diagram. They show the correctness of their estimator under some assumptions. In Chazal et al. (2025), topological signatures for periodic signals are defined (each time series is assigned a signature time series that should be independent of reparametrisations). They study the additivity property of diagrams of sublevel sets of periodic functions defined on any intervals, and prove convergence and stability results for their signatures. The additivity property is also leveraged in Bois et al. (2022) to study multiple sclerosis through motion sensor time series: barcodes from sublevel sets are truncated to reduce the impact of the number of steps, then the bottleneck distance is used in UMAP (McInnes et al., 2018) to globally analyze a cohort of healthy subjects and patients with different degree of severity of the disease.

**Featurization and integration in learning pipelines.** Several studies, mostly with biomedical applications, use the sublevel set filtration with vectorizations of persistence diagrams used as inputs for machine learning or statistics. Importantly, such pipelines have been applied to EEG time series, resulting in several notable studies published between 2018 and 2020. A prominent approach involves using permutation-based statistical tests on persistence landscapes to detect statistically significant differences between time series. This methodology has been applied to the study of epileptic seizures (Wang et al., 2018; 2019) and later aphasia (Wang et al., 2020b;a), by comparing time series recorded before and after a seizure, or between pathological and healthy subjects. More precisely, in Wang et al. (2018; 2019), Fourier series approximations of the EEGs are computed, and the test compares persistence landscapes obtained when exchanging Fourier coefficients between time series. The authors empirically show that the test they defined is invariant to translations, scaling, and frequency scaling and claim that it detects topological changes. Also note that Wang et al. (2020a) introduces the gradient filtration, which generalises the sublevel set filtration by allowing level lines with a non-zero slope. Alternatively, Piangerelli et al. (2018) uses persistent entropy (Chintakunta et al., 2015) as a feature for classifiers to analyze EEGs in order to detect epileptic seizures. A similar approach was later applied in Majumder et al. (2020) to detect autism spectrum disorders. In Nasrin et al. (2019), a Bayesian learning approach is developed with Poisson point processes as prior for the distribution of persistence diagrams, with an application to brain state classification from EEGs. More recently, sublevel sets have also been used for heart rate variability detection in segmented ECG signals (i.e., signals where heartbeats are isolated) in Graff et al. (2021), using various features describing the persistence diagram in an SVM classifier.

Deep learning architectures have also been designed to leverage properties of sublevel sets filtrations. Betti curves (Dindin et al., 2020) are fed to 1D convolutions in neural networks (Umeda (2017) claimed that 1D convolutions are suitable for Betti curves) and added to non-TDA features to improve arrhythmia detection and classification in ECG data. Zeng et al. (2021) aimed to integrate local topological information on time series by using sublevel filtrations of subwindows and learnable barcode vectorizations (Carrière et al., 2020; Hofer et al., 2019b;a) to design a topological attention mechanism (Vaswani et al., 2017) to improve state-of-the-art time series forecasting neural networks.

**Transformation of the time series.** All the above methods perform a direct analysis of graphs of time series. Another approach consists of transforming the time series into a more relevant form, such as converting it to the frequency domain as did Chen et al. (2019); Chen & Ravishanker (2023), who then used the Euclidean distance between persistence landscapes from sublevel set filtrations into a k-means clustering algorithm. In Chen et al. (2019), the Walsh-Fourier Transform (a frequency domain transform adapted to categorical time series) is applied to categorical time series (with few possible states), which transforms it into a real-valued time series that is more suitable for signal processing tools such as sublevel set filtrations. In Chen & Ravishanker (2023), the filtration is applied to the smoothed periodogram (a Fourier-based transform that estimates the spectral density of a signal) to study environmental time series. A related approach that does not use the sublevel set filtration but the filtered cubical complex (Kaczynski et al., 2006), a related filtration used for image processing, is described in Reise et al. (2024). The authors filter the spectrogram (Fourier-based 2D transform of a time series) of subwindows and use the $L^1$ distance between Betti curves in 0 and 1D, followed by a minimum-cost matching approach to perform audio signals identification, with empirical evidence of robustness to several types of signal obfuscations (noise, reverb, filtering, tempo/pitch shift). Finally, another relevant method to directly filter a time series' graph is introduced in Dlugas (2022) to detect P,Q,S, or T-waves in ECG signals. The authors study the graph of the ECG (as a subset of $\mathbb{R}^2$) to which they add the x-axis to form loops (1-cycles) that are detected using the Rips filtration and volume-optimal representatives (Obayashi, 2018). Each cycle is identified with a kind of ECG wave according to criteria based on medical knowledge.

Table 2 sums up the characteristics of all the articles cited in this section.

Table 2: Articles from Section 4.2 and their characteristics.

| Article | Year | Field | Time series | Transformation | Filtration | TDA tools | Method | Theory |
|---|---|---|---|---|---|---|---|---|
| Khasawneh & Munch (2018) | 2018 | Industry | Binary signals with noise | Truncated time series | Sublevel (x-axis) | Barcode | Count bars above threshold | Model + correctness guarantees |
| Wang et al. (2018) | 2018 | Biomedical | EEG | Fourier series approximation | Sublevel | PL | Permutation test | No |
| Piangerelli et al. (2018) | 2018 | Biomedical | EEG | Filtering, downsampling | Sublevel (VR second step) | Barcode + entropy (generators) | Linear classifier (histograms) | No |
| Wang et al. (2019) | 2019 | Biomedical | EEG | Fourier series approximation | Sublevel | PL | Permutation test | No |
| Nasrin et al. (2019) | 2019 | Biomedical | EEG | None | Sublevel | PD | Bayesian learning | No |
| Chen et al. (2019) | 2019 | Geographic | Categorical | Walsh-Fourier transform | Sublevel | PL | K-means | No |
| Dindin et al. (2020) | 2020 | Biomedical | ECG | None | Sublevel | Betti curve | Deep learning | No |
| Wang et al. (2020b) | 2020 | Biomedical | EEG | Average over trials | Sublevel | PL | Permutation test | No |
| Wang et al. (2020a) | 2020 | Biomedical | EEG | Average over trials | Sublevel (gradient) | PL | Permutation test | Introduces gradient filtration |
| Majumder et al. (2020) | 2020 | Biomedical | EEG | None | Sublevel | Barcode + entropy | SVM | No |
| Myers et al. (2020) | 2020 | Generic | Univariate | None | Sublevel | PD | Thresholding | Full statistical study |
| Zeng et al. (2021) | 2021 | Generic | Univariate | Subwindows | Sublevel | Barcode + learned repr. | Deep learning | No |
| Graff et al. (2021) | 2021 | Biomedical | ECG | Segmentation | Sublevel | Features from PD | SVM | No |
| Bois et al. (2022) | 2022 | Biomedical | Motion sensors | None | Sublevel | Truncated barcode + bottleneck | UMAP | No |
| Dlugas (2022) | 2022 | Biomedical | ECG | Add x-axis | VR | 1-cycle extraction | Feature analysis | No |
| Chen & Ravishanker (2023) | 2023 | Ecological | Univariate | Periodogram | Sublevel | PL | K-means | No |
| Tanweer et al. (2024) | 2024 | Generic | Univariate | Truncated time series | Sublevel (x-axis) | Barcode | Intervals from thresholded PDs | No |
| Bonis et al. (2024) | 2024 | Industry | Magnetic field | Function on circle | Sublevel | PD | Counting-based estimator | Theoretical properties |
| Reise et al. (2024) | 2024 | Audio | Musical signals | Spectrogram | Filtered cubical complex | Betti curve $(L^1)$ | Matching | No |
| Chazal et al. (2025) | 2025 | Generic | Periodic | None | Sublevel | Topological signatures | None | New signatures + theory |

### 4.3 Delay embedding methods

For a univariate time series $(x_t)_{1 \le t \le n}$, the subwindow of length $l < n$ of a time series $x$ is a sequence of the form $(x[i], x[i+1], \dots, x[i+l-1])$. The delay embedding of $x$ with dimension $d \in \mathbb{N}$ and delay $\tau \in \mathbb{N}$ is the following point cloud in $\mathbb{R}^d$:

$$X_{d,\tau} = (X_{d,\tau}[t])_{1 \le t \le n-(d-1)\tau}$$
$$= ((x[t], x[t+\tau], \dots, x[t+(d-1)\tau]))_{1 \le t \le n-(d-1)\tau}$$

Figure 6 shows a univariate time series (a sum of sinusoidal functions with period 2) and its delay embedding in dimension $d = 2$ with delay $\tau = 5$. The delay embedding is made of two concentric ellipses: the small one corresponds to the smallest oscillations of the signal (roughly between -1 and 1) and the large one corresponds to the largest oscillations (roughly between -3 and 3).

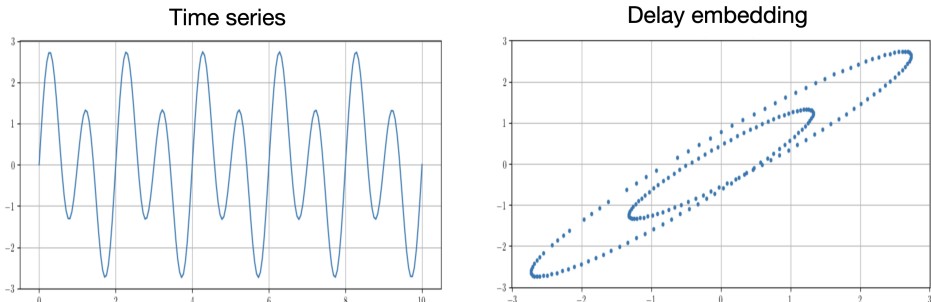

Figure 6: **Left:** A univariate time series: $x[t] = \sin(\pi t) + 2\sin(2\pi t)$. **Right:** the corresponding delay embedding with $d = 2$ and $\tau = 5$.

Delay embeddings are a popular transformation of time series, originally introduced to study dynamical systems. Takens' theorem (Takens, 2006) states that a smooth attractor (a manifold of dimension $m$) of a dynamical system can be reconstructed from a delay embedding of an observation function, provided that the embedding dimension is at least $2m + 1$. More generally, points of a delay embedding represent local variations of the time series; in particular, if the delay is $\tau = 1$, each point corresponds to a subsequence of length $d$, forming a point cloud in $\mathbb{R}^d$. This point cloud can be analyzed with various tools, including persistent homology in dimensions 0 to $d - 1$, typically via persistence diagrams computed from the Vietoris–Rips filtration of the delay embedding, as presented in Section 3.

Combining delay embeddings with persistent homology has led to numerous studies, demonstrating the relevance of this approach for a wide variety of tasks and application domains. Figure 7 illustrates a typical pipeline for coupling delay embeddings with persistent homology tools. We now review some key thematic connections related to this approach.

**Identifying periodic behavior.** We have already presented in Section 4.2 several approaches for analyzing periodic time series using sublevel sets. An alternative and historically older line of work, encompassing a larger body of research, focuses instead on studying the one-dimensional persistent homology of a delay embedding of the time series. An important early theoretical contribution in this direction is the work of Perea & Harer (2015), which provides a comprehensive theoretical study of the one-dimensional persistent homology of the Rips filtration of delay embeddings of periodic functions. In particular, they derived a lower bound for maximum persistence of Fourier series approximations (with $N$ frequencies) and their limits as $N \to \infty$, when parameters $d$ and $\tau$ satisfy $d\tau = T$ and $d = 2N$ ($T$ being the period of the function). The idea is that the delay embedding of a continuous function is a closed curve, so it has at least one persistent feature in its 1D persistent (singular) homology, so the same should happen from time series that are samplings of periodic functions.

Building on these ideas, maximal persistence in one-dimensional persistent homology has been used to quantify periodicity in various applications, such as detecting periodic patterns in gene expression time series (Perea et al., 2015). It is also worth noting that the use of one-dimensional maximal persistence from

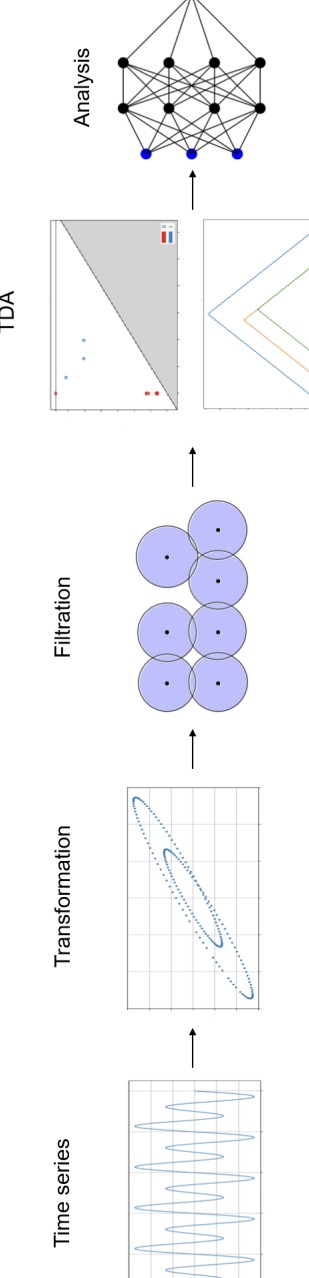

Figure 7: Typical framework of persistence homology for time series using a delay embedding.

a delay embedding was proposed earlier in Emrani et al. (2014) for wheeze detection in breathing sound signals, predating the publications of Perea & Harer (2015) and Perea et al. (2015). In that work, the authors introduce a model for wheeze signals and show that, with an appropriate choice of delay embedding parameters, the Rips filtration produces at least one persistent feature in the one-dimensional persistence diagram, a feature that is absent in non-wheeze signals. Similarly, Khasawneh & Munch (2016) employed the maximal one-dimensional persistence of the Rips filtration of delay embeddings for chatter detection in synthetic dynamical systems with industrial applications. This work was extended in Khasawneh et al. (2018), introducing new features for both 0D and 1D homology, including maximal persistence and polynomial features from Adcock et al. (2016), which are independent of points with zero persistence, and applying them within a machine learning framework using a logistic regression classifier.

The results of Perea & Harer (2015) were extended to quasiperiodic functions (sums of complex-valued trigonometric functions whose frequencies are linearly independent over $\mathbb{Q}$) in Perea (2016) and very recently in Gakhar & Perea (2024). In particular, it was shown that under some hypotheses, the delay embedding of a quasiperiodic function with $N$ frequencies is dense in an $N$-torus embedded in $\mathbb{C}^{d+1}$. The authors of Gakhar & Perea (2024) also derive lower bounds for the number of persistent features in dimension 1 to $N$ and for their persistence. They show that their work can be used to detect dissonance in music.

Another method for periodicity detection and period estimation was proposed by Dłotko et al. (2019), also based on 1D persistent homology. They use the Čech filtration and total $p$-persistence ($p$th root of the sum of all persistence values to the power $p$) to detect periodicity, then a symbolization-based method for period estimation (note that the algorithm and choice of parameters are not explicitly described).

**Identifying chaotic behavior.** Closely related to the previous works, an emerging line of work is dedicated to distinguishing chaotic from periodic behavior in time series using persistence diagrams (or vectorizations of persistence diagrams) from delay embeddings. An important seminal work by Ichinomiya is Ichinomiya (2023), later continued in Ichinomiya (2025). The proposed approach is based on recurrence plots (see, e.g., Marwan et al., 2007), a popular and powerful tool for analyzing complex dynamical systems. The idea is to first perform a delay embedding, then compute the distance matrix of the resulting point cloud, and finally compute the 0-degree persistence diagram associated with the sublevel sets of this distance matrix (viewed as a grayscale image). This approach has the advantage of a computational complexity of order $O(N^2)$, where $N$ denotes the number of points in the embedded cloud, regardless of the embedding dimension. In the first paper (Ichinomiya, 2023), the author provides empirical evidence that this method can qualitatively distinguish between chaotic and periodic behaviors from persistence diagrams. In Ichinomiya (2025), the approach is further applied to the clustering of electromyograms (EMGs) and the classification of electrocardiograms (ECGs). In this second work, the persistence diagrams are transformed into persistence images (PIs), and the dimension is then reduced via nonnegative matrix factorization (NMF) before applying standard clustering and classification techniques. In a similar direction, the recent work of Shah et al. (2025) provides further insights into the link between persistent homology and important parameters governing chaotic time series. More precisely, the authors show how CROCKER plots (Topaz et al., 2015), Betti curves, and persistence landscapes obtained from delay embeddings can be used to visualize and infer key information about chaotic phase transitions.

**Change point detection and anomaly detection.** Delay embeddings can be used to detect changes in time series by studying topological changes in the embeddings of subwindows. Note that a drawback is that this requires many computations of persistence diagrams and bottleneck distances. Ueda et al. (2022) proposed a change point detection method (for financial data) based on this idea: they defined statistical models of persistence diagrams and use them to compute a sequence of features from a sequence of persistence diagrams corresponding to successive subwindows. Existing change point detection methods can then be applied to those features. Fernández et al. (2023) studied the persistent homology of point clouds using Fermat distances. The idea is that in order to recover topological properties of a manifold embedded in $\mathbb{R}^d$, Fermat distances are less dependent on the manifold embedding than the Euclidean distance, and this applies to delay embeddings. The authors propose a method for time series analysis (best suited for change point detection): they study delay embedding of subwindows starting at 0 and of increasing lengths, compute the Rips filtration with the Fermat distance, which gives a sequence of persistence diagrams. The sequence of bottleneck distances between consecutive diagrams has peaks that correspond to change points.

This was only applied to a few examples, but the method comes with theoretical guarantees (convergence, robustness).

Closely related to change point detection is the problem of anomaly detection. In a recent work (Bois et al., 2024a), the anomaly detection problem for univariate time series is formalized using a convolutional model: the studied time series are successions of patterns that are normal (i.e., frequent) or abnormal. The above ideas are also relevant to this model. The authors defined an anomaly detection method along with a theoretical analysis. They leverage properties of DTM filtrations (Anai et al., 2020) (robustness to outliers, possibility of subsampling a point cloud while integrating density information from the whole point cloud) to identify 1-cycles that correspond to normal patterns, and points far away from them are detected as anomalies.

**Computational improvements.** Computation times for persistent homology of point clouds can be reduced by using dimension reduction (Kim et al., 2018; Majumdar & Laha, 2020), filtrations that have fewer vertices than there are points in the point cloud (Fraser et al., 2017; Sanderson et al., 2017; Yesilli et al., 2022; Bois et al., 2022), or zigzag persistence (Tymochko et al., 2020). Zigzag persistence is an extension of persistent homology to non-increasing sets of simplicial complexes (Carlsson & De Silva, 2010). Kim et al. (2018) proposed a generic featurization algorithm for time series, which they applied to financial data. They use delay embeddings and principal component analysis (PCA) for dimension reduction, Rips complexes, and persistent landscapes and silhouettes as inputs for deep learning models. They derived stability theorems with respect to noise, sampling, and PCA, under the assumption that, for $l$-dimensional PCA, the point cloud lies in an $l$-dimensional linear subspace, and the PCA matrix $X^\top X$ has at most $l$ positive eigenvalues. They also discuss how PCA can help denoise while preserving topology (under some assumptions). Majumdar & Laha (2020) use PCA on delay embeddings of subwindows to save computation time when computing on the persistent landscape per subwindow. They studied financial time series using self-organising maps for clustering and random forests for classification. Beyond the techniques based on PCA, it should also be noted that the methods developed in the previously mentioned works Ichinomiya (2023) and Ichinomiya (2025) can also be regarded as dimensionality reduction techniques. The approach based on the sublevel sets of the distance matrix significantly reduces computational complexity and can be applied generally, beyond the specific contexts for which it was originally designed.

Fraser et al. (2017) proposed an ECG visualization method based on delay embeddings of subwindows and for several delay values. Witness complexes (De Silva & Carlsson, 2004) are used to save computation time and memory usage. They are complexes constructed using only a subset of the points as vertices. The output is a 2D plot of the first Betti number of some simplicial complex (not detailed) as a function of time and delay. Sanderson et al. (2017) also use the witness complex to perform binary classification of musical signals using a feature computed from the 1D persistent rank function (Robins & Turner, 2016).

In an update of Khasawneh et al. (2018), Yesilli et al. (2022) use Bézier curves fitted on the delay embedding to save computation time for the Rips filtration and persistent homology (method from Tsuji & Aihara, 2019, about 30 times faster). They experiment with several featurization methods: features from Khasawneh et al. (2018), landscapes (Bubenik & Dłotko, 2017), persistent images (Adams et al., 2017), a kernel method (Reininghaus et al., 2015), and persistence paths signatures (Chevyrev et al., 2018), and several classifiers to classify real industrial data. In the work described above, Bois et al. (2024a) use a subsampled point cloud but integrate density information from the whole point cloud into their DTM filtration.

Tymochko et al. (2020) use zigzag persistence (Carlsson & De Silva, 2010) to detect bifurcations (qualitative changes in a system's behavior) in dynamical systems. Their data is a sequence of time series (typically, a parameterized dynamical system). They fix a radius and compute all Rips complexes for this radius. Then zigzag persistence is applied to the obtained sequence of simplicial complexes. They identify the appearance and disappearance of bifurcations as the birth and death times of persistent points on the 1D persistence diagrams. One advantage over other approaches is that only one persistence diagram is computed instead of one for each time series, but fixing the radius is a drawback.

A recent article (Martinez et al., 2025) proposes a method to approximate the birth–death pairs of $H_1$-persistent features in 2D point clouds without computing simplicial complexes, relying instead on fitting ellipses that represent large-scale $H_1$ features. Applied to synthetic data consisting of non-stationary time series, the method is shown to accurately approximate persistence diagrams while greatly reducing computation time (about 1900 times). An interesting research direction is to extend it beyond two

dimensions.

**Featurization and integration in learning pipelines.** Following our previous remarks on featurization, several studies have explored alternative representations of persistent homology, often with the goal of integrating them into learning pipelines. In particular, in the study of biomedical time series such as ECG (Liu et al., 2023; Safarbali & Golpayegani, 2019; Jiang et al., 2022; Ling et al., 2022; Mjahad et al., 2022; Ren et al., 2023; Ignacio et al., 2019; 2020), motion sensor data (Yan et al., 2020b;a; 2022; Tong et al., 2021) or financial data (Gidea et al., 2020; Aguilar & Ensor, 2020; Guo et al., 2020; Zhang & Wu, 2025; Guritanu et al., 2025). Applications include classifying diseases, detecting critical events (arrhythmia, fibrillation, financial crisis), or finding different regimes. The general method consists of computing persistent homology of the Rips or Čech filtration of a delay embedding (maybe after some preprocessing), then using vectorized persistence diagrams or features from persistence diagrams as input for classifiers (SVM, random forest, deep learning) or k-means clustering. See Table 3 for more details about the application of each article and specific choices of vectorization and classifiers/clustering method. In this context, a specific line of work, initiated by Umeda (2017), relies on the use of Betti curves. They showed that Betti curves obtained from Čech filtrations of delay embeddings are sensitive to scale changes in the time series. They claim that 1D convolutional neural networks are suitable to learn from Betti curves and use them in neural networks to improve performance in time series classification on biomedical data. Delay embeddings and Betti curves were used for other biomedical applications in later works. Yamanashi et al. (2021) performed detection of delirium on bispectral EEG using the area under the 1D Betti curve as a score for a statistical test. Yan et al. (2022) used features from Betti curves, persistent landscapes, and silhouettes in several classifiers to detect freezing-of-gait episodes in motion sensor data (due to Parkinson's disease).

In recent work, the cross-barcode feature map has been proposed, together with a novel discrepancy measures known as the Manifold Topology Divergence (MTD) and Representation Topology Divergence (RTD) (Barannikov et al., 2021; 2022; Mironenko et al., 2026). The cross-barcode encodes the topological relationship between two filtrations directly into a single persistent object, thereby providing an alternative to the standard pipeline, which consists in computing two persistence diagrams separately and subsequently comparing them via bottleneck or Wasserstein distances. More precisely, Barannikov et al. (2021) introduced the MTD and cross-barcodes to compare two point clouds embedded in the same ambient space. Subsequently, Barannikov et al. (2022) introduced the RTD and (R)-cross-barcodes to compare two point clouds equipped with a point-to-point correspondence, possibly lying in different ambient spaces. Finally, Mironenko et al. (2026) developed a statistical theory of cross-persistence diagrams and barcodes, including density results, for both types of cross-barcode constructions. Applications include the study of training dynamics of GANs on time series data, in particular financial data (Barannikov et al., 2021), as well as gravitational wave detection and time series classification (Mironenko et al., 2026).

**Choosing delay embedding parameters.** The choice of parameters $d$ and $\tau$ can have a major impact on the performance of delay embedding-based methods. When studying continuous periodic functions, theory (Perea & Harer, 2015) suggests choosing $d$ as high as possible, with the window size $d\tau$ equal to the period (the fixed window size is also supported by empirical results from Bois et al., 2024a). A high-dimensional embedding with a small delay will finely represent the local variations of the function. However, there are several problems in practice. Firstly, working in high dimensions can be computationally expensive. Secondly, "irrelevance" and "redundancy" (Tan et al., 2023) must be avoided. Irrelevance happens when $d\tau$ is too large, and the coordinates of each point of the embedding are too far away. In this case, the embedding fails to capture local variations, and a point can be influenced by events far apart in time, making interpretation difficult. As $\tau$ cannot be chosen arbitrarily small in practice, this implies that $d$ should not be too high in practice. This also highlights the importance of signal sampling quality. Redundancy, on the other hand, happens when $\tau$ is too small, so variations are slow between consecutive points of the time series, leading to point clouds that are concentrated on the diagonal $x_1 = x_2 \cdots = x_d$, making it difficult to identify persistent features, especially in the presence of noise.

Tan et al. (2023) review the main classical methods for selecting the delay $\tau$ and embedding dimension $d$. These include dynamical approaches such as the false nearest neighbors method (Kennel & Abarbanel, 2002), the quarter-of-period heuristic (Judd & Mees, 1998), and criteria based on autocorrelation or mutual

information (see, e.g., Kantz & Schreiber, 2003), as well as more "geometric" methods such as fill-factor (Buzug & Pfister, 1992) and noise amplification (Uzal et al., 2011). They also provide an overview of popular techniques for selecting non-uniform time lags. In addition, they introduce the SToPS (Significant Times on Persistent Strands) method, which constructs delay-embedding point clouds for candidate lags and computes persistent homology to track the emergence of topological features. This produces a characteristic time spectrum that highlights lags best capturing the underlying dynamics. SToPS can be used to select both uniform and non-uniform delays, and is shown to be competitive with the classical approaches mentioned above. Alternatively, Myers et al. (2024) propose guidelines for choosing $d$ and a topological method for selecting $\tau$ in order to compute the permutation entropy (Bandt & Pompe, 2002), a useful measure of the complexity of the signal. Their approach identifies the maximum "significant" frequency present in the time series (following the idea developed in Melosik & Marszalek, 2016), which they approximate via the 0-th persistence diagram of the sublevel sets of the time series, either in the time or frequency domain. Empirically, they show that this method reliably identifies delays that capture the underlying dynamics (for various classical performance measures), performing comparably to or better than traditional heuristic procedures across periodic, chaotic, and more complex time series.

Note that these methods are heuristics that are not guaranteed to perform well in every application. For example, they can fail in the presence of noise or with non-periodic signals or for non-uniform samplings. Moreover, complex methods can have parameters that are themselves hard to choose. Specific knowledge about the application can help define better and/or simpler heuristics.

To overcome these calibration difficulties, some works, most notably Tran & Hasegawa (2018), depart from the usual approach of fixing $d$ and $\tau$, and instead develop delay-variant methods that allow these parameters to vary. Such methods reveal multi-timescale patterns in a time series by enabling the observation of variations in topological features, with the time delay acting as an additional dimension in the topological feature space. In practice, this can be implemented by computing persistence diagrams (or related summaries) for various values of $\tau$ and/or $d$. In Tran & Hasegawa (2018), the authors demonstrate, in a time-series classification framework, that these delay-variant approaches outperform methods based on selecting a single pair $(\tau, d)$, and more generally compare favorably with standard time-series analysis techniques. However, these methods come with a significantly higher computational cost.

**Other delay embedding-based transformations.** A few articles studying univariate time series use transformations that are not exactly delay embeddings but have similarities. Germain et al. (2024) proposed an algorithm for motif discovery in time series (a motif is a pattern that appears several times in the time series). They use the LT-normalized distance (Germain et al., 2023) (distance invariant to linear trend) to construct a weighted k-nearest-neighbors graph (k-nearest non-overlapping subsequences) with additional temporal connections between successive points. Motifs are then reconstructed using a 0D persistence-based clustering algorithm (Bois et al., 2024b). Although the set of subsequences can be seen as a delay embedding with delay 1, only the distances to the non-overlapping k nearest neighbors are taken into account for persistent homology (which saves computation time), and time is taken into account with temporal connections, which is significantly different than a Rips filtration on a delay embedding. Temporal connections were also used by Venkataraman et al. (2016) for action recognition on motion capture data (note that they have multivariate data but study each dimension independently). They connected successive points of the delay embedding, before computing the Rips filtration and using a nearest-neighbor classifier on persistence diagrams endowed with the 1-Wasserstein distance.

In a series of works, Myers et al. (2019; 2022; 2023), use the ordinal partition network (McCullough et al., 2015) to transform time series into unweighted graphs. This transformation takes a delay embedding of parameters $d, \tau$ and associates a permutation $\pi$ of $[1, d]$ to each point $X_t = (x_t, x_{t+\tau,...,x_{t+(d-1)\tau}}) = (X_1, X_2, \ldots, X_d)$ (the permutation that sorts the $d$ coordinates, i.e. such that $X_{\pi(1)} < X_{\pi(2)} < ... < X_{\pi(d)}$). The set of permutations obtained is the set of vertices of the graph, and an edge goes from $i$ to $j$ if there exists a time $t$ such that permutations $i$ and $j$ respectively correspond to $X_t$ and $X_{t+1}$. The graph can then be filtered using the Rips filtration from the shortest path distance (weighted or not) or diffusion distance. In Myers et al. (2019), the periodicity score from Perea et al. (2015) is extended to unweighted graphs, and it is used along with other features from 1D persistent homology, such as persistent entropy (Chintakunta et al., 2015), to distinguish periodic behavior from

chaotic behavior in dynamical systems. In Myers et al. (2022), the bottleneck distance between persistence diagrams in multidimensional scaling is used, and then dynamic state detection in dynamical systems is performed with an SVM classifier. The method proposed in Myers et al. (2023) in the case of univariate time series (dynamical systems with intermittency, i.e., irregular transitions from periodic to chaotic dynamics) computes a sequence of ordinal partition networks from subwindows and zigzag persistence. Then they detect intermittency using the coordinates of the most persistent points on the 1D persistence diagram. The multivariate applications of this method are described in Section 4.4.

Note that many of the methods presented in this section are designed for uniformly sampled data and may yield unsatisfactory results when applied to irregularly sampled time series. In particular, irregular sampling can introduce spurious topological features in the reconstructed state space. To address this issue, a recent work (Dakurah & Cisewski-Kehe, 2024) proposes a novel embedding approach specifically tailored for irregularly sampled time series, based on the extraction of uniformly spaced subsequences. This embedding is proven to preserve the topology of the original state space, reduce spurious features, and be robust to noise. Additionally, the authors demonstrate that their method outperforms standard time-delay embedding techniques on several irregularly sampled examples involving both synthetic and real datasets.

Table 3: Articles from Section 4.3 and their characteristics.

| Article | Year | Field | Time series | Transformation | Filtration | TDA tools | Method | Theory |
|---|---|---|---|---|---|---|---|---|
| Emrani et al. (2014) | 2014 | Biomedical | Breathing sounds | Delay embedding | VR | PD | Max 1D persistence | Model and theoretical study |
| Perea & Harer (2015) | 2015 | Generic | Periodic | Delay embedding | VR | PD | Max 1D persistence | Full theoretical study |
| Perea et al. (2015) | 2015 | Biomedical | Gene expression time series | Delay embedding | VR | PD | Max 1D persistence | No |
| Perea (2016), Gakhar & Perea (2024) | 2016 & 2024 | Audio signal processing | Quasiperiodic | Delay embedding | VR | PD | Max persistence | Full theoretical study |
| Khasawneh & Munch (2016) | 2016 | Industry | Dynamical systems | Delay embedding | VR | PD | Visualization | No |
| Venkataraman et al. (2016) | 2016 | Motion capture | 3D position time series | Delay embedding | VR + temporal connections | PD | KNN classifier | No |
| Umeda (2017) | 2017 | Biomedical | Motion sensor data, ECG, EMG | Delay embedding | Čech | Betti curve | DL | Introduction of Betti curves |
| Sanderson et al. (2017) | 2017 | Music | Music time series | Delay embedding | Witness | Persistent rank function | Binary classification | No |
| Fraser et al. (2017) | 2017 | Biomedical | ECG | Delay embedding of subwindows | Witness | Betti number | Visualization | No |
| Khasawneh et al. (2018) | 2018 | Industry | Dynamical systems | Delay embedding | VR | 0D and 1D max pers. | Machine learning | No |
| Kim et al. (2018) | 2018 | Generic | Synthetic, financial | Delay embedding then PCA | VR | PL, silhouette | DL | Stability with respect to noise, sampling and PCA |
| Tran & Hasegawa (2018) | 2019 | Generic | Biological | Delay embedding | VR | PD for varying $\tau$ | Kernel methods | No |
| Dlotko et al. (2019) | 2019 | Generic | Periodic | Delay embedding | Čech | Total persistence | Symbolization | No |
| Myers et al. (2019) | 2019 | Dynamical systems | Dynamical systems | Delay embedding then unweighted graph | VR | Barcode | Features from 1D PH | No |
| Safarbali & Golpayegani (2019) | 2019 | Biomedical | ECG | Delay embedding | Čech | Average 1D persistence | DL | No |
| Ignacio et al. (2019) | 2019 | Biomedical | ECG | Delay embedding | VR | Features from barcode | Random forest classifier | No |
| Ignacio et al. (2020) | 2020 | Biomedical | ECG | Delay embedding of subwindows | VR | Features from barcode and PL | Random forest classifier | No |
| Gidea et al. (2020) | 2020 | Financial | Log-return of cryptocurrencies | Delay embedding of subwindows | VR | PL | K-means clustering | No |
| Majumdar & Laha (2020) | 2020 | Financial | Stocks | Delay embedding of subwindows and PCA | VR | PL | Self-organizing maps (DL) or random forest classifier | No |
| Tymochko et al. (2020) | 2020 | Dynamical systems | Sequence of time series | Delay embeddings and VR complexes | Zigzag | PD | Most persistent points | No |
| Yan et al. (2020b) | 2020 | Biomedical | Motion sensor data | Delay embedding | VR | PL | Random forest classifier | No |
| Yan et al. (2020a) | 2020 | Biomedical | Motion sensor data | Delay embedding | VR | PL | Various classifiers | No |
| Aguilar & Ensor (2020) | 2020 | Financial | Stocks returns | Delay embedding | VR | PL | Permutation tests | No |

| Article | Year | Field | Time series | Transformation | Filtration | TDA tools | Method | Theory |
|---|---|---|---|---|---|---|---|---|
| Guo et al. (2020) | 2020 | Financial | Stocks returns | Delay embedding | VR | PL | Visualization | No |
| Barannikov et al. (2021; 2022) | 2021 & 2022 | Financial | Stocks returns | N.A. | VR | Cross-barcode | Study of training dynamics of GANs | Stability and structural properties |
| Yamanashi et al. (2021) | 2021 | Biomedical | EEG | Delay embedding | Čech | Betti curve | Area under the 1D Betti curve | No |
| Tong et al. (2021) | 2021 | Biomedical | Gait features | Delay embedding | Čech | Persistent entropy | SVM classifier | No |
| Myers et al. (2022) | 2022 | Dynamical systems | Dynamical systems | Graph | VR | PD | MDS and SVM | No |
| Yesilli et al. (2022) | 2022 | Industry | Dynamical systems | Delay embedding and Bézier curves | VR | PL, max pers., or other featurization methods | Machine learning | No |
| Ueda et al. (2022) | 2022 | Financial | Time series with change points | Delay embedding of subwindows | Čech | PD | Statistical model on PDs, feature, change point detection algorithm. | NML codelength derivation |
| Yan et al. (2022) | 2022 | Biomedical | Motion sensor data | Delay embedding | VR | PL, Betti curve, persistent silhouette | Various classifiers | No |
| Ling et al. (2022) | 2022 | Biomedical | ECG | Moving average + delay embedding | VR | Features from PD | Various classifiers | No |
| Mjahad et al. (2022) | 2022 | Biomedical | ECG | Filtering + delay embedding | VR | PD, PL, persistent silhouette | KNN classifier | No |
| Jiang et al. (2022) | 2022 | Biomedical | BCG | Delay embedding | VR | Features from barcode | Various classifiers | No |
| Liu et al. (2023) | 2023 | Biomedical | ECG | Segmentation and delay embedding | Čech | PL | Random forest classifier | No |
| Myers et al. (2023) | 2023 | Various | Dynamical systems with intermittency | Ordinal partition of subwindows | Zigzag | PD | Most persistent points | No |
| Ren et al. (2023) | 2023 | Biomedical | ECG | Delay embedding | VR and sublevel | Barcode | DL | No |
| Fernández et al. (2023) | 2023 | Generic | Any | Delay embedding | VR | PD of subwindows | Distance between consecutive PDs | Theoretical analysis |
| Ichinomiya (2023) | 2023 | Generic | Dynamical systems | Delay embedding then recurrence plots | sublevel on distance matrix | PD | Visualization | No |
| Tan et al. (2023) | 2023 | Generic | Dynamical systems | Delay embedding | VR | PD | SToPS | No |
| Germain et al. (2024) | 2024 | Generic | Alternating motifs | KNN Graph | VR + temporal connections | PD | Persistence-based clustering | No |
| Bois et al. (2024a) | 2024 | Generic | Normal or abnormal patterns | Delay embedding | DTM Rips | PD and 1-cycles extraction | Distance to normal cycles | No |
| Myers et al. (2024) | 2024 | Generic | Dynamical systems | Delay embedding | VR and sublevel | PD | State space reconstruction and Permutation Entropy | No |
| Dakurah & Cisewski-Kehe (2024) | 2024 | Generic | Irregularly sampled | Subsequence embedding | VR | PD | State space reconstruction | Stability results and convergence analysis |
| Ichinomiya (2025) | 2025 | Generic | EMG and ECG | Delay embedding then recurrence plots | sublevel on distance matrix | PI | NMF and SVM classifier | No |

| Article | Year | Field | Time series | Transformation | Filtration | TDA tools | Method | Theory |
|---|---|---|---|---|---|---|---|---|
| Shah et al. (2025) | 2025 | Generic | Dynamical systems | Delay embedding | VR | CROCKER plot, PL, Betti curve | Visualization | link between PH and Chaos-related parameters |
| Zhang & Wu (2025) | 2025 | Financial | Stocks returns | Delay embedding | VR | PL | Random Forest classifier | No |
| Guritanu et al. (2025) | 2025 | Financial | Stocks returns | Delay embedding | VR | PL | Causal decision rule | No |
| Martinez et al. (2025) | 2025 | Generic | non-stationnary time series | Delay embedding | None | Approximated $H_1-$PD | Visualization | No |
| Mironenko et al. (2026) | 2026 | Generic | Any | Delay embedding | VR | Cross-Barcode and MTD | Classification, event detection | Statistical fondations |

### 4.4 Multivariate time series

In this section, we review applications of persistent homology to multivariate time series. A straightforward way to extend the tools presented in the previous sections to this setting is to compute persistent-homology descriptors separately for each channel. Several studies have adopted this strategy. Here, however, we focus on approaches that explicitly account for the multivariate structure of the data. Figure 8 illustrates a typical framework for the analysis of multivariate time series.

**Graph-based methods.** Beyond applications of channel-wise methods, the most popular way to study multivariate time series is to transform them into an undirected weighted graph whose vertices correspond to each channel of the time series and edge weights represent a distance or similarity between them. The graph is then filtered using a clique filtration (ascending for distances, descending for similarity measurements) to study vectorized persistence diagrams.

Graph-based methods are mainly applied to study brain signals (mostly EEG or fMRI) through a notion of coherence, dependency, or correlation, which is motivated by the fact that neurons that are wired together in the brain tend to have correlated activations. An early and influential application of graph-based topological methods in neuroscience is the work of Petri et al. (2014). They introduced a new object for topological data analysis: the homological scaffold (HS), which is a graph whose vertices are the ones from the original graph and whose edge weights encode information about the number or persistence of the homology generators containing each edge. They studied the effect of psychedelic mushrooms on the brain with fMRI data. They built a correlation graph from time series and used a descending clique filtration to compute persistence diagrams and homological scaffolds. They use scaffolds to visualize brain connectivity and also analyze statistical features from persistence diagrams and homological scaffolds.

Later, Stolz et al. (2017; 2021) studied multivariate fMRI time series to study motor learning in healthy subjects and schizophrenia. In their first article (Stolz et al., 2017), they use a functional connectivity measure (coherence of the wavelet scale-2 coefficients), a descending clique filtration, and they interpret changes in the persistence landscapes of different days in terms of synchronization. They insist on the fact that low-persistence features can contain important information. In the second article (Stolz et al., 2021), they use Pearson correlations, a descending clique filtration, and compute persistence landscapes (and average persistence landscapes) and images for clustering and statistical tests.

More recently, El-Yaagoubi et al. (2023a;b) studied EEG time series from ADHD patients. They transform their data into graphs using Fourier-based distances (respectively, dependency-based distance and frequency-specific distance) and use ascending clique filtrations and statistical tests on persistence landscapes. The test used in El-Yaagoubi et al. (2023a) is a permutation test from Robinson & Turner (2017) based on the $L^2$ distance between landscapes to compare EEGs of healthy subjects and patients with ADHD. In the second article (El-Yaagoubi et al., 2023b), they introduce a hypothesis test based on persistence landscapes and derive associated convergence results. Manjunath et al. (2023) used the same frequency-specific distance, clique filtration, and permutation test on EEGs to study obstructive sleep apnea. El Yaagoubi & Ombao (2023) also studied EEG signals by transforming them into a directed graph to study epileptic seizures. Their goal is to integrate the directed flow of information between brain regions. First, they use a non-symmetric distance function called the partial directed coherence, which is a Granger causality-based dependence measure, to get a distance matrix. A distance is then defined from the anti-symmetric part of the weight matrix, which is used to compute the directed graph and its clique filtration, and persistence diagram. Diagrams are compared before and after seizures.

Interestingly, a very recent study by Bhattacharya et al. (2025) compares a graph-based approach with the channel-wise featurization of persistent homology introduced in Aithal et al. (2024). In the context of mild cognitive impairment (MCI) classification from fMRI data, the authors compare two models: a CNN classifier that takes as input a matrix of pairwise distances between the persistence diagrams of each channel (obtained via delay embedding and the construction of the corresponding Vietoris–Rips filtration), and a stacked ensemble classifier that uses as input the persistence diagrams derived from the clique filtration of the correlation graph between channels. In their case study, the channel-wise approach outperforms the graph-based one. This result highlights that constructing a correlation graph may lead to a loss of important information, likely because the correlation graph is a static representation of dynamic interactions.

Beyond biomedical data, studying sequences of local correlation networks has also proven to be useful to

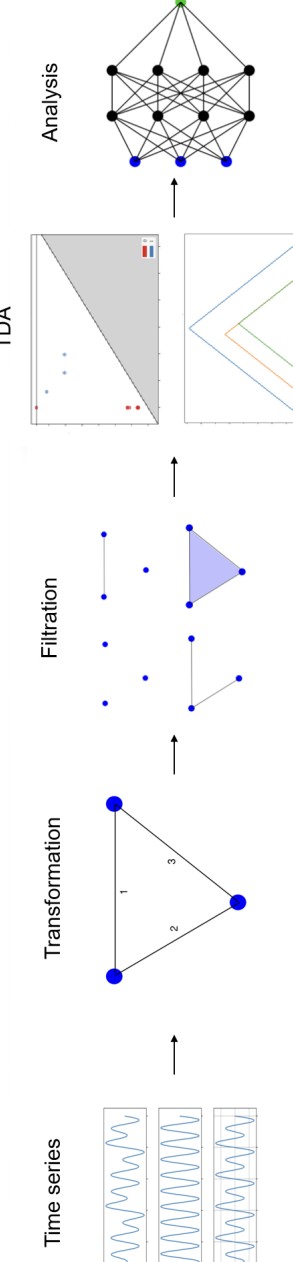

Figure 8: Typical framework of persistence homology for multivariate time series.

study stock markets (Gidea, 2017) or detect multivariate anomalies (Chazal et al., 2024). Both works use a correlation-based graph transformation on subwindows and clique filtrations to obtain a time series of persistence diagrams. Gidea (2017) studies the sequence of 2-Wasserstein distance between each diagram and the initial one to detect critical transitions in financial markets. Chazal et al. (2024) introduce an algorithm for anomaly detection in multivariate time series. They vectorize the obtained persistence diagrams using a measure quantization algorithm (Royer et al., 2021), then estimate the mean and covariance matrix of the time series of vectorized diagrams. This mean and covariance matrix represent the normal behavior. An anomaly score is defined based on the Mahalanobis distance (i.e., the normalized distance from each vectorized diagram to the mean of the normal model). They give a theoretical study of several aspects of their method.

**Point cloud-based methods.** Multivariate time series with $d$ channels can be seen as time-indexed point clouds in $\mathbb{R}^d$. This makes it possible to use usual persistent homology methods such as those from Section 4.3. As for delay embeddings of univariate time series, 1D persistent homology can be used to detect periodic behaviors in multivariate time series. A seminal example is the work of Vejdemo et al. (2015), which studies periodic motion with position and orientation time series for several body parts. They see the $d$-variate time series as a point cloud in $\mathbb{R}^d$, compute its Rips filtration and 1D persistence barcode, then parameterize the point cloud with circular coordinates for each persistent bar using a method from De Silva & Vejdemo-Johansson (2009). They use a clustering technique to compute typical trajectories and interpolate between different periodic motions while preserving periodicity. Later, Tralie & Perea (2018) extended the delay embedding to multivariate data as a transformation from $\mathbb{R}^{W \times H}$ to $\mathbb{R}^{d \times W \times H}$ ($W \times H$ is the number of pixels), and defined (quasi-) periodicity scores based on maximal 1D and 2D persistence following ideas from Perea & Harer (2015); Perea (2016); Gakhar & Perea (2024), presented in the previous section. They apply their method to (quasi-) periodicity quantification in videos (time series of images).
Gidea & Katz (2018) studied multivariate financial time series by seeing subwindows of length $w$ as point clouds of size $w$ in $\mathbb{R}^d$. They use the Rips filtration and compute the $L^p$ norm of 1D persistence landscapes (for $p = 1, 2$) as a function of time. They studied this new univariate time series using its variance and the average spectral density at low frequencies, and gave empirical evidence that they increase before a financial crisis.
In a methodological paper, Salch et al. (2021) study multivariate fMRI data as a time-varying function on voxels $f : (x, y, z, t) \mapsto f(x, y, z, t)$ (i.e., a time series for each voxel). For each time $t$, they compute the Čech filtration of the point cloud in $\mathbb{R}^4$ made of all points $(x, y, z, f(x, y, z, t))$. They propose to study the corresponding 1D persistence vineyard (Cohen-Steiner et al., 2006), i.e., the time series of 1D persistence diagrams, with a notion of spatial proximity between features at time $t$ and $t + 1$. They use a statistical test and visualization method to identify and study task-responsive structures. Note that this technique could also be used in higher dimensions. A drawback is that vineyards are not robust as the temporal connection between features depends on their representatives (Cohen-Steiner et al., 2006). The authors indicate that using random representatives or specific ones could help improve the method.

**Other methods.** Santoro et al. (2023) generalize the notion of correlation between two time series to any number $k$ of time series by introducing the $k$-order co-fluctuation and using to define a simplicial complex where, for all $k \geq 2$, each $(k - 1)$-simplex represents a group of $k$ channels of the multivariate time series and is associated to the corresponding $k$-order co-fluctuation value. Then, the complex is filtered by adding simplices by decreasing values, excluding those that do not respect the filtration condition (which are labeled as violations). The persistent homology generators and violations are used to define features (hyper-coherence and hyper-complexity) describing the persistence diagrams and homological scaffolds. They show the relevance of their features for various types of multivariate time series (fMRI, financial, and historical data of infectious diseases).
Zigzag persistence has also been applied to multivariate time series by Corcoran & Jones (2017) and more recently by Myers et al. (2023). Corcoran and Jones analyze fish swarm behavior (multi-agent trajectories over time) using zigzag persistence on a sequence of simplicial complexes built from upper-level sets of a fixed-threshold kernel density estimator. They then compute persistence landscapes and apply K-medoids clustering with the $L^2$ distance. The method of Myers et al. (2023) targets time-varying graphs and thus extends naturally to multivariate time series via standard graph constructions.

Table 4: Articles from Section 4.4 and their characteristics.

| Article | Year | Field | Time series | Transformation | Filtration | TDA tools | Method | Theory |
|---|---|---|---|---|---|---|---|---|
| Petri et al. (2014) | 2014 | Biomedical | Multivariate fMRi | Weighted graph (correlations between time series) | Clique (descending) | PD and HS | Statistical features from PDs and HSs, and visualisation with HS | Introduction of HS |
| Vejdemo et al. (2015) | 2015 | Motion capture | Position and orientation time series | Point cloud | VR | 1D barcode and circular coordinates | Clustering | No |
| Corcoran & Jones (2017) | 2017 | Biology | Position time series | Upper-level set of kernel density estimator | Zigzag | PL | K-medoid clustering (L2 distance) | No |
| Stolz et al. (2017) | 2017 | Biomedical | Multivariate fMRi | Weighted graph (functional connectivity) | Clique (descending) | PL | Visualization, K-means | No |
| Gidea (2017) | 2017 | Finance | Stock returns | Weighted graph (correlation-based distance) | Clique (cending) | PD (as-) | Distance to initial PD | No |
| Gidea & Katz (2018) | 2018 | Finance | Stock returns | Point cloud from subwindows | VR | $L^p$ norm of PLs | Variance and the average spectral density | No |
| Tralie & Perea (2018) | 2018 | Video processing | Videos | Multivariate delay embedding, normalization, SVD | VR | PD | Scores based on max 1D and 2D persistence | No |
| Stolz et al. (2021) | 2021 | Biomedical | Multivariate fMRi | Weighted graph (correlation) | Clique (descending) | 1D PL, persistence image (de-) | Clustering, classification, statistical test | SVM No |
| Salch et al. (2021) | 2021 | Biomedical | Multivariate fMRi | Point cloud | Čech | Persistence vineyard | Visualization, statistical test | No |
| El-Yaagoubi et al. (2023a) | 2023 | Biomedical | Multivariate EEG | Weighted graph (dependency-based distance based on Fourier coefficients) | Clique (cending) | PL (as-) | Permutation test (based on $L^2$ distance between PLs) | No |
| Manjunath et al. (2023) | 2023 | Biomedical | Multivariate EEG | Weighted graph (dependency-based distance based on Fourier coefficients) | Clique (cending) | PL (as-) | Permutation test (based on $L^2$ distance between PLs) | No |
| El-Yaagoubi et al. (2023b) | 2023 | Biomedical | Multivariate EEG | Weighted graph (dependency-based distance based on Fourier coefficients) | Clique (cending) | Spectral PLs (a PL for each frequency) (as-) | Hypothesis testing | Definition of a test statistic on spectral landscapes and associated convergence results |
| Santoro et al. (2023) | 2023 | Various | Multivariate (fMRI, financial, historical data of infectious diseases) | Normalization of each channel, then simplicial complex (k-order co-fluctuation) | Decreasing k-order fluctuation, with violations excluded | PD and HS + violating simplices | Features (hyper-coherence and hyper-complexity) | No |
| El Yaagoubi & Ombao (2023) | 2023 | Biomedical | Multivariate EEG | Weighted directed graph (partial directed coherence) | VR | PD | Visualization | No |
| Chazal et al. (2024) | 2024 | Generic | Multivariate with anomalies | Graph (correlation-based distance) | Clique (cending) | PD and vectorization (measure quantization) (as-) | Mahalanobis distance to normal model | Theoretical analysis |
| Bhattacharya et al. (2025) | 2025 | Biomedical | Multivariate fMRi | Graph (correlation-based distance) | Clique (cending) | PD (as-) | stacked ensemble classifier | No |

## 5 Discussion and perspectives

Throughout this survey, we have highlighted the usefulness of persistent homology for a variety of tasks in time series analysis, with applications spanning multiple fields. To synthesize these contributions, we propose in Figure 9 a visual taxonomy that enables the reader to quickly navigate the thematic structure of the reviewed works. We close with concluding remarks on the strengths and limitations of persistent homology, highlight some literature gaps, and outline perspectives for its use in time series analysis.

**The value of persistent homology.** Time series analysis has a long history, with the development of efficient and well-understood tools grounded in well-established disciplines such as nonparametric statistics, functional analysis, dynamical systems, and graph theory, among others. In this context, one may naturally wonder about the value and relevance of methods based on persistent homology. The relevance of persistent homology lies in its ability to capture the multi-scale geometric and topological structure of complex data, revealing patterns (at different scales) that are often overlooked or obscured by conventional statistical summaries. By translating the evolving topology of a signal into persistence diagrams, it produces interpretable and geometrically grounded summaries. Further, as presented in Section 3, persistence diagrams and related representations are stable under small perturbations or noise. This combination of stability and interpretability makes persistent homology a distinctive tool for time series analysis, bridging geometry and traditional statistical methodology. Yet, practitioners should be aware that persistent homology and its standard representations are inherently invariant under several transformations of the data (for instance, reparametrization). While these invariances can be interesting in certain contexts, they also imply that some information is lost. Consequently, persistent homology is not universally the most appropriate tool for every problem, and in many applications, it should be better used as a complement to more classical statistical or machine learning methods. This point is often overlooked in works proposing TDA-based tools for time-series analysis, which generally do not provide theoretical or empirical comparisons with classical methods. We believe this is unfortunate, as such comparisons would help clarify the added value of these new tools and thereby enhance their impact and adoption among practitioners.

**Which method for which use ?** A key practical question is: for a given task, which class of methods should one choose ? In the case of univariate time series, when are sublevel-set–based approaches preferable to delay-embedding techniques ? This is a subtle issue, and no definitive answer exists. Nevertheless, the present survey provides some guidance. For instance, in tasks related to topological filtering, such as detecting extrema or zero crossings, sublevel-set methods tend to be more appropriate. More broadly, in settings where interpretability is important, working directly with persistent homology on the raw signal offers clear advantages. By contrast, for problems such as identifying chaotic dynamics or detecting anomalies, the literature has largely focused on approaches based on delay embeddings. There is, however, a substantial overlap between these two paradigms, where both can perform competitively. This is notably the case for identifying periodic or quasi-periodic behavior, as well as for designing feature representations that integrate persistent homology into machine learning or deep learning pipelines. In practice, the choice is often application-dependent, and practitioners are encouraged to experiment with methods from both perspectives.

**Interpretability, large and small scale features.** As illustrated by several of the works we reviewed, representations of persistent homology can often be interpreted in a relatively straightforward way and linked to meaningful dynamical or structural properties, such as local extrema (Myers et al., 2020) and zero-crossings (Tanweer et al., 2024), periodicity (Khasawneh & Munch, 2018; Bonis et al., 2024; Chazal et al., 2025), or chaos-related indicators (Ichinomiya, 2023; Shah et al., 2025), which are valuable in many fields, notably biomedicine. However, many existing approaches use persistent homology merely as a feature extractor for machine-learning pipelines, which tends to obscure these interpretative advantages. Moreover, methods that rely directly on persistent-homology representations have the additional benefit of being fully unsupervised, unlike most machine-learning-based techniques. Developing such methods, therefore, appears to be an important direction for the future of the TDA research community. One other appealing aspect of persistent homology is its genuinely multi-scale nature. Unlike many descriptors that capture features of the data only at a fixed scale, persistent homology aggregates information about

homological structures across all scales, providing a comprehensive view of their evolution. However, most existing work on persistence-based time-series analysis tends to focus exclusively on the most prominent features (those corresponding, in some sense, to large-scale structures), typically by discarding points in the persistence diagram that lie close to the diagonal. In many applications, such small-scale features are treated as noise, which is often justified in practical settings such as those cited previously. Yet, recent studies have shown that small-scale features can carry meaningful information. For instance, Bubenik et al. (2020a) demonstrate, using toy examples, that points near the diagonal in persistence diagrams obtained from sampled point clouds can be used to infer the curvature of the support of the underlying distribution. Similarly, in his thesis, Perez (2022) shows that small bars in persistence diagrams of sublevel sets of stochastic processes can be exploited to infer regularity and self-similarity parameters governing these processes. These results offer new insights into the interpretation of persistence representations and open the door to promising applications (particularly in the context of time-series analysis) that, to the best of our knowledge, have not yet been explored. Further theoretical progress in understanding the information contained in small-scale features would also be highly beneficial for future developments and applications.

**Delay embedding parameters**. Points in delay embeddings capture local variations of a time series, while persistent homology provides tools to study their global organization through various filtrations and methodological choices. However, as highlighted above, the embedding dimension and delay parameters have a substantial influence on the resulting topology and on its subsequent interpretation. Existing procedures for selecting these parameters are largely heuristic and often introduce additional hyperparameters. Developing approaches that are less sensitive to these choices or that adaptively select them remains an important challenge. Intrinsic metrics, such as the one proposed in Fernández et al. (2023), offer a promising direction in this regard. More broadly, identifying an appropriate notion of distance for constructing filtrations is crucial for fully exploiting the potential of TDA in time-series analysis.

On the other hand, methods that do not rely on a fixed choice of delay-embedding parameters, but instead treat them as additional variables, such as in Tran & Hasegawa (2018), are appealing, yet suffer from high computational cost. One interesting possibility would be, rather than computing a separate persistence diagram for each pair $(\tau, d)$, to consider $\tau$ and/or $d$ as additional filtration parameters and compute a global topological descriptor associated with this multi-dimensional filtration, ideally, at a much lower computational cost. This falls within the scope of multiparameter persistent homology, an emerging topic that is briefly discussed below.

Additionally, existing delay-embedding approaches tend to be restricted to low-dimensional persistent features (typically connected components and, at most, one-dimensional loops). Developing methods (and theory) that reliably produce, interpret, and leverage higher-dimensional persistent features could therefore open new perspectives.

**Beyond correlation.** For multivariate data, we have highlighted that a substantial portion of the literature focuses on studying the topology of correlation graphs. As previously mentioned, this approach can overlook important information and may, in some situations, be outperformed by simpler persistence-based methods (Bhattacharya et al., 2025). A promising direction for future research is therefore to move beyond correlation networks by incorporating richer structures such as $k$-order co-fluctuation patterns (Santoro et al., 2023), directed (causality-based) graph transforms (El Yaagoubi & Ombao, 2023), or even persistence vineyards (Salch et al., 2021). These tools have the potential to provide a more nuanced characterization of dynamical interactions and could, in particular, offer valuable insights into brain function through improved analyses of EEG and fMRI data.

**Implementation and computation time.** We should highlight that the TDA community has made substantial efforts to develop accessible implementations of its core tools, such as those listed in Table 5. Nevertheless, the computational cost of persistent homology continues to be a significant limitation. For instance, in the case of Vietoris–Rips filtrations, a dataset of $n$ points generates on the order of $O(n^{k+1})$ simplices when truncated at homological dimension $k$. Since the standard matrix reduction algorithm scales cubically with the number of simplices, the worst-case time complexity for computing the $k$-th persistence diagram grows as $O(n^{3(k+1)})$. This severe combinatorial explosion constitutes the primary computational bottleneck, and becomes particularly prohibitive for large datasets or data embedded in

high-dimensional ambient spaces. Several advances have been proposed to mitigate this cost. Notably, optimized implementations of persistent homology for Rips filtrations (Bauer, 2021; Zhang et al., 2020; Pérez et al., 2021) significantly improve practical performance, while linear-size approximation schemes (Sheehy, 2012) reduce the complexity of the underlying complexes. In addition, as discussed in Section 4.3, strategies such as dimensionality reduction and the use of sparse complexes can further alleviate computational burden. Despite these developments, scalability remains a central challenge, and further progress in this direction would substantially broaden the applicability of persistent homology.

| Library | Language | Link |
|---------|----------|------|
| GUDHI | C++, Python, R | `https://gudhi.inria.fr/` |
| Dionysus | C++ | `https://www.mrzv.org/software/dionysus/` |
| DIPHA | C++ | `https://github.com/DIPHA/dipha` |
| Giotto | Python | `https://giotto-ai.github.io/gtda-docs/0.4.0/library.html` |
| PHAT | C++ | `https://bitbucket.org/phat-code/phat/src/master/` |

Table 5: Libraries for topological data analysis

**Persistence parameters.** In some of the reviewed methods, two (or more) parameters are relevant to construct a filtered simplicial complex. For example, distance and time (Tymochko et al., 2020), density and time (Corcoran & Jones, 2017), distance and density (Bois et al., 2024a) or distance and delay-embedding parameters (Tran & Hasegawa, 2018). In the first two cases, a distance/density parameter is fixed and zigzag persistence is used on a time-varying simplicial complex. In the third case, density information is integrated into the Rips filtration by the DTM filtration (Anai et al., 2020), with a fixed parameter (number of neighbors). In the fourth case, persistence diagrams are computed for embeddings corresponding to multiple values of the parameter $\tau$. Vasudevan et al. (2011) also use several thresholds for sublevel sets but do not specify how they define the filtration, even though it is non-trivial.

A way to study the structure of data across all values of several parameters is multiparameter persistent homology (Botnan & Lesnick, 2022). However, it is difficult to interpret because multiparameter persistence modules do not have an interval-decomposition (the decomposition that gives the persistence diagram for usual persistence). There exist invariants and vectorizations for multiparameter persistent homology (usually based on 1-parameter reductions), and some are starting to be applied to time series classification (Loiseaux et al., 2024; Kim & Jung, 2022; 2024). We did not include multiparameter persistence in the body of this review because this is still a mainly theoretical research topic that would require a lot only to introduce a few applications. Nevertheless, it is a topic that readers should be aware of, as it offers important perspectives for future developments. Dupont (2020)

Figure 9: Taxonomy of the 87 reviewed works. The blocks correpond to each thematic paragraphs from Sections 4.2, 4.3 and 4.4.

**Acknowledgments**

This research was supported by the DATAIA Convergence Institute as part of the Investment for the Future Program (ANR-17-CONV-0003) at ENS Paris-Saclay.

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
