# OpenReview forum: "Persistent homology for time series: a selective review"
_TMLR — Accepted by TMLR_

### Review · Reviewer_TJai · 2026-03-02

**Summary Of Contributions:**

This paper presents a comprehensive and well-structured selective review of the use of persistent homology for time series analysis. The authors survey 84 papers published between 2014 and 2025 across a wide range of application domains, including biomedicine, dynamical systems, signal processing and etc.

The review introduces a unifying conceptual pipeline:

time series → transformation → filtration → TDA tools → analysis

which is used consistently to categorize and compare existing methods. The paper provides clear mathematical background on persistent homology and filtrations, followed by a systematic taxonomy of approaches based on sublevel set filtrations, delay embeddings, graph-based methods, and multivariate extensions.

Key strengths:
- Broad and up-to-date coverage of the relevant literature.
- A clear unifying framework that helps organize and compare existing methods.

Weaknesses:
- The review is mostly descriptive; a deeper critical comparison between method families would strengthen the paper.
- Computational constraints of the presented methods are not discussed in details.

**Audience:**

Yes

**Audience Explanation:**

Yes. The paper provides a useful, up-to-date overview of persistent homology for time series and offers a clear organizational framework that is relevant to both TDA and ML researchers.

**Broader Impact Concerns:**

The paper is a methodological review and raises no direct ethical concerns.

**Claims And Evidence:**

Yes

**Claims Explanation:**

The claims are supported by a broad and relevant set of references across application domains and method families. The proposed framework is well aligned with the surveyed methods and consistently illustrated throughout the paper. The technical background is accurate and appropriate for a review article.

**Requested Changes:**

Critical:

- Briefly clarify the paper selection process (e.g., search strategy and inclusion criteria) to improve transparency and reproducibility of the review.

- While the paper provides a clear taxonomy of method families, a slightly more explicit comparative discussion (e.g., when sublevel-set vs. delay-embedding approaches are preferable in practice) would further strengthen the review.

Non-critical:

- Include a short discussion of computational cost and scalability.

---

### Review · Reviewer_u1ci · 2026-03-04

**Summary Of Contributions:**

The paper proposes a review on persistent homology applied to time series analysis tasks, covering also its application as a featurization technique for downstream machine learning models. Overall the review spans both recent and past contributions, systematically identifying the characteristics of different works (e.g., application domain, type of time series, techniques adopted). The review also introduces the theoretical background to understand persistent homology and concludes with a discussion about potential pitfalls and future research directions.

**Strengths:**
- The paper is well written
- Concepts are formally introduced, allowing an unfamiliar reader to understand what persistent homology is about.
- The literature review is extensive, covering the last decades.
- It can effectively serve as an introduction to persistent homology techniques to a reader mostly familiar with machine learning/deep learning techniques.

**Weaknesses:**
- While well written, the theoretical introduction can be hard to parse, as multiple concepts are introduced in rapid sequence and the provided examples might be unclear at times.
- The literature review is very broad, however the paper does not go in depth on any particular method.
- The reviewed works are not explicitly organized in a taxonomy, this makes it hard to revisit the paper to look at a glance for relevant works in the area one might be interested in.

**Audience:**

Yes

**Audience Explanation:**

I think persistent homology might be new to many TMLR readers. While recent deep learning approaches in time series moved further and further away from "handcrafted" featurization, discovering methodologies such as the one presented in the paper can be interesting and spark new ideas. As such, I think this work can be interesting as an introduction to persistent homology for some readers. Nonetheless, it might not be strongly relevant in the current deep learning for time series landscape.

**Broader Impact Concerns:**

The work is a literature review, I don't have any broader impact concerns.

**Claims And Evidence:**

Yes

**Claims Explanation:**

The paper sets the goal of "summarizing the main trends in the field and provide readers with a clear and comprehensive overview of the key ideas and methods developed to date". I think it achieves its goal as it introduces the relevant concepts and covers a broad range of previous works, providing for each some information about the approach and application area.

**Requested Changes:**

First I will list some major suggestions, mainly regarding how I think the work can be improved in its effort of introducing a machine learning/deep learning researcher to the presented concepts. Then there are some minor changes (typos etc.). Note that "major" here just serves to separate the degree of importance. It is not in the sense of "necessary".

**Major:**
- Sec 3 could be made easier to follow by introducing an example (like Fig. 1) but already centered around time series. In that case as new definitions are introduced, the reader could be shown how the corresponding items can be calculated in the case of the example.
- Figure 1 introduces a "persistence diagram". However, it is not explained what that is until later on. I would move the persistence diagram to appear later, maybe using Figure 3 to show both persistence diagram and barcodes relating to Figure 1.
- Figure 3 refers to a figure which is 7 pages ahead. If Figure 3 showed the persistence diagram and barcode for Figure 1 it would be easier to follow through.
- While the review is very extensive, it lacks depth on any particular method. Maybe highlighting some important methods for each considered category, possibly showing some practical examples, can guide better the reader into which works might be more relevant to read first.
- Introducing a visual taxonomy of the different methods can make it easier for a reader to quickly identify the works he might be more interested in. The current table-based approach, while providing further information, can be hard to parse. After reading the paper it would be useful to have something to quickly scan to see, e.g., works related to anomaly detection on univariate series.

**Minor:**
- Title can be updated to use standard uppercase formatting.
- Hyperlinks to tables, figures, and references should be introduced.
- Tables style can be improve to match standard publication formatting used in ML
- Table 1 could use the term "Agnostic" or "General" instead of "All".
- Second line, second paragraph, sec 3.3: should it be K^{\alpha^i}?
- Line 5, last paragraph, page 16. Typo: should be "of features" instead of "a features"
- End of first paragraph of page 18: dangling "Outside"

---

### Review · Reviewer_KSv1 · 2026-04-08

**Summary Of Contributions:**

This paper provides a comprehensive review of the persistent homology for time series data, covering 84 studies. It describes the basic concepts of topological data analysis (TDA) and explains how persistent homology captures structural features. The authors consider a unified framework that decomposes existing procedure into a pipeline consisting of transformation, filtration, topological analysis, and some subsequent target tasks. Two main approaches are highlighted: direct analysis using sublevel set filtrations and geometric analysis via delay embeddings. The review surveys a wide range of applications, particularly in biomedicine, as well as in industrial systems, finance, and signal processing. The paper also discusses challenges, including computational complexity, parameter selection, and the need for more theoretical and empirical validation.


Strength

- According to Table 1, this survey provides a significantly more comprehensive coverage compared to existing reviews on TDA for time series.

- It systematically organizes TDA-based approaches across a wide range of tasks (e.g., filtering, analysis of periodic data, feature extraction, etc.), offering a broad overview of the field.

Weakness

- The mathematical descriptions are not easily accessible to readers without prior familiarity with TDA, and in some parts, the presentation becomes a rather monotonic listing of concepts.

- The survey is largely descriptive and lacks quantitative empirical evaluation. In the current era of large language models, it is unclear how much value such purely enumerative reviews provide.

- There is very limited discussion of implementation aspects, which may reduce the practical usefulness of the paper for practitioners.

**Audience:**

Yes

**Audience Explanation:**

Topological data analysis should be a topic that attracts interests from TMLR's audience.

**Claims And Evidence:**

Yes

**Claims Explanation:**

As the authors claim, the paper seemingly provides comprehensive review of time series TDA.

**Requested Changes:**

I think readability of mathematical descriptions should be improved, for which I felt difficulty to follow, while I do not have an other severe concern throughout the paper.

Regarding Section 3, while the authors provide a mathematical foundation for persistent homology, I found the current presentation is difficult to follow for practitioners who are not familiar with topological data analysis (TDA). The current presentation in Section 3 suffers from an abrupt transition to high-level abstraction. Section 3 introduces several formal definitions (e.g., simplicial complexes, filtrations, chain complexes, homology groups) in a highly abstract and algebraic manner, but provides limited intuitive explanations or concrete examples.

The transition from simplicial homology (Section 3.2) to persistent homology (Section 3.3) is abrupt. While boundary operators, chain groups, and homology groups are defined, there is little explanation of how these concepts connect operationally to the computation of persistence diagrams. In particular, the key idea, tracking the 'birth' and 'death' during a filtration, is not sufficiently motivated and explained before introducing diagram and barcode (actually, the words 'birth' and 'death' are not explicitly defined, though these are essential keywords of TDA).

Section 3 introduces multiple types of filtrations (sublevel sets, Cech, Rips, clique filtrations) without clearly explaining when and why each should be used in the context of time series analysis. This results in a catalog of definitions rather than a guided explanation.

The quality and clarity of several figures could be improved, as they are currently somewhat rough and may confuse readers. For example,
- Figure 3: The caption states that this figure corresponds to Figure 6, but the relationship between the two is not clearly explained. It is unclear how the persistence diagram in Figure 3 is constructed from the data in Figure 6. Moreover, it seems to rely on concepts (sublevel set filtrations) that have not yet been introduced at that point in the paper, which may be confusing for readers.
- Figure 4: The figure lacks axis labels and a legend, making it difficult to interpret. In addition, the scale of the horizontal axis does not appear to be consistent with that of Figure 3a.
- Figure 5: The red lines in the filtration diagram are not explained in the text or caption. Additionally, the power spectral density appears in the figure without any explanation.

While a variety of problem settings are discussed in section 4, it is not always clearly revealed what limitations of non-TDA approaches motivate the use of TDA, or what can be uniquely achieved by adopting TDA. In many cases, there are few or no references to alternative non-TDA methods, making it difficult to understand the relative positioning of TDA with respect to existing approaches (non-TDA) for each task.

If possible, the paper would benefit from a clearer discussion of implementation aspects. While many methods are described conceptually, there is little guidance on practical implementation, such as available software libraries. Including such details would improve the usability of the survey for practitioners.

Minor comment:
- In page 18, before 'Choosing delay ...', 'Outside'.

---

### Author Response · Authors · 2026-04-20
**Revision (1/3)**

Dear Editor and Reviewers,

We sincerely thank you for your thoughtful and insightful comments and suggestions. They have been extremely helpful in improving the quality and clarity of our manuscript. Below, we provide a detailed, point-by-point response to each of your comments. All corresponding revisions have been carefully incorporated into the manuscript and are highlighted in green in the revised version for ease of reference.

- **Sec 3 could be made easier to follow by introducing an example (like Fig. 1) but already centered around time series. In that case as new definitions are introduced, the reader could be shown how the corresponding items can be calculated in the case of the example.**

The example presented in Figure 1 has been replaced with a simpler illustration that is more directly related to time series, namely the sublevel set filtration of a univariate time series.

- **Figure 1 introduces a "persistence diagram". However, it is not explained what that is until later on. I would move the persistence diagram to appear later, maybe using Figure 3 to show both persistence diagram and barcodes relating to Figure 1**

This is a good remark. The persistence diagram has been removed from Figure 1 so that the figure focuses solely on the filtration. The corresponding persistence diagram (and barcode) are presented later in Figure 3.

- **Figure 3 refers to a figure which is 7 pages ahead. If Figure 3 showed the persistence diagram and barcode for Figure 1 it would be easier to follow through.**

This is also a good remark. Following the previous comment, the persistence diagram and barcode in Figure 3 now correspond to the filtration introduced in Figure 1 rather than the one from Figure 6.

- **While the review is very extensive, it lacks depth on any particular method. Maybe highlighting some important methods for each considered category, possibly showing some practical examples, can guide better the reader into which works might be more relevant to read first.**

Although this is a fair point, the primary motivation behind this survey is to provide an informal yet broad overview of applications of persistent homology to time series analysis. In contrast to existing surveys (see Table 1), which typically focus on a specific class of methods (or a limited subset) our aim is to encompass a much wider range of approaches and contributions. Given the substantial diversity of the selected papers, both in terms of tasks and methodological tools, emphasizing particular methods in greater detail would risk overshadowing others that may be equally relevant. For this reason, we have chosen to focus more extensively on the underlying ingredients (Section 3) rather than on the individual methods themselves.

- **Introducing a visual taxonomy of the different methods can make it easier for a reader to quickly identify the works he might be more interested in. The current table-based approach, while providing further information, can be hard to parse. After reading the paper it would be useful to have something to quickly scan to see, e.g., works related to anomaly detection on univariate series**

This is a great idea! We have added a visual taxonomy at the end of the Discussion section that organizes the reviewed works into blocks, each corresponding to a paragraph in Sections 4.1–4.3.

---

### Author Response · Authors · 2026-04-20
**Revision (2/3)**

- **Briefly clarify the paper selection process (e.g., search strategy and inclusion criteria) to improve transparency and reproducibility of the review.**

The present survey is intended as a selective rather than a systematic review, and as such, the selection process is not fully reproducible. It was conducted in two stages. First, an initial corpus of papers was identified by a Google Scholar search using the query “persistent homology time series”. Second, this corpus was expanded by incorporating additional works that either cite or are cited by these initial papers, particularly when they highlight important aspects not captured by the original query. We have added a brief paragraph in the Introduction to clarify this point.

- **While the paper provides a clear taxonomy of method families, a slightly more explicit comparative discussion (e.g., when sublevel-set vs. delay-embedding approaches are preferable in practice) would further strengthen the review.**

This is indeed an important practical question, but a delicate one, as it is often highly data-dependent. Moreover, the literature rarely provides direct comparisons between these different types of methods. We have therefore added a paragraph in the Discussion section that offers some elements of guidance on this issue; however, a definitive answer remains out of reach.

- **Include a short discussion of computational cost and scalability.**

A paragraph discussing this issue was already present in the discussion section, which we have extended to more clearly highlight the computational limitations of persistent homology tools. Additionally, section 4.2 includes a dedicated paragraph presenting several computational improvements.

---

### Author Response · Authors · 2026-04-20
**Revision (3/3)**

- **Regarding Section 3, while the authors provide a mathematical foundation for persistent homology, I found the current presentation is difficult to follow for practitioners who are not familiar with topological data analysis (TDA). The current presentation in Section 3 suffers from an abrupt transition to high-level abstraction. Section 3 introduces several formal definitions (e.g., simplicial complexes, filtrations, chain complexes, homology groups) in a highly abstract and algebraic manner, but provides limited intuitive explanations or concrete examples.**

We agree with the reviewers’ comment that the previous draft introduced key notions from algebraic topology too abruptly and in a manner that was overly formal. In the revised version, we have therefore added more intuitive explanations to help motivate and clarify these concepts before their formal presentation.

- **The transition from simplicial homology (Section 3.2) to persistent homology (Section 3.3) is abrupt. While boundary operators, chain groups, and homology groups are defined, there is little explanation of how these concepts connect operationally to the computation of persistence diagrams. In particular, the key idea, tracking the 'birth' and 'death' during a filtration, is not sufficiently motivated and explained before introducing diagram and barcode (actually, the words 'birth' and 'death' are not explicitly defined, though these are essential keywords of TDA).**

Similarly, we include a short paragraph at the beginning of Section 3.3 that provides an informal motivation for persistent homology, emphasizing its connection to classical homology in an intuitive manner:

“Persistent homology studies how the homology groups of a space evolve as the filtration parameter increases. It captures the appearance and disappearance of topological features across scales, producing a discrete summary known as a persistence diagram (or equivalently, barcodes). In this summary, each feature is represented by its \textbf{birth}, i.e the time at which it appears (e.g., a new connected component emerges or a loop forms) and its \textbf{death}, i.e. the time at which it disappears (e.g., two connected components merge or a loop is filled). This provides a compact, scale-invariant representation of the topological structure of the data.”*

 We also introduce formal definitions of the birth and death times of topological features, which were previously omitted, in order to clarify their interpretation within the persistence framework.

- **Section 3 introduces multiple types of filtrations (sublevel sets, Cech, Rips, clique filtrations) without clearly explaining when and why each should be used in the context of time series analysis. This results in a catalog of definitions rather than a guided explanation.**

Regarding this comment, we explain when each type of filtration is typically used by explicitly referring to the corresponding subsections of Section 4. In particular, we emphasize that the sublevel set filtration can be applied directly to univariate time series (Section 4.1), while Čech and Rips filtrations are used on point clouds obtained via delay embedding of time series (Section 4.2). Finally, we note that the clique filtration is applied to covariance or correlation graphs derived from multivariate time series (Section 4.3)

- **Figure 5: The red lines in the filtration diagram are not explained in the text or caption. Additionally, the power spectral density appears in the figure without any explanation.**

The red lines shown in the figures are intended to represent the sublevel set filtration, corresponding to the levels at which topological features emerge. The power spectral density is merely provided as an example of a possible transformation applied to the time series prior to constructing the sublevel set filtration. If this leads to confusion, we can remove the label “power spectral density” to improve clarity.

- **While a variety of problem settings are discussed in section 4, it is not always clearly revealed what limitations of non-TDA approaches motivate the use of TDA, or what can be uniquely achieved by adopting TDA. In many cases, there are few or no references to alternative non-TDA methods, making it difficult to understand the relative positioning of TDA with respect to existing approaches (non-TDA) for each task.**

This is a fair point. However, it is delicate to draw a direct comparison between TDA and non-TDA methods, as very few papers in the considered corpus include such comparisons with non-TDA baselines. This limitation is indeed discussed in the paragraph entitled “The value of persistent homology” in the Discussion section.

---

### Decision · Action_Editor_ALKo · 2026-05-29

**Recommendation:** Accept with minor revision

**Additional Comments:**

Decision: Minor Revision

The reviewers agree that the manuscript provides a broad, useful, and well-organized survey of persistent homology methods for time-series analysis, and that the authors have addressed the main concerns raised during review.

I share the reviewers’ view that the paper satisfies the TMLR acceptance criteria as a survey contribution. The main remaining limitation is that the article is largely descriptive: it provides relatively limited critical comparison between method families and only modest practical guidance on when particular persistent-homology pipelines should be preferred. Nevertheless, this limitation does not preclude publication in TMLR.

Before acceptance, I request a **minor revision** addressing the following points:

First, the 1994 work [1] should be cited in Section 3.3 where the manuscript first introduces persistence diagrams/barcodes as the discrete birth–death summary associated with a filtration, and again where it discusses the computation of persistent pairs and persistence barcodes. Section 3.4 should also mention [1], where stability of persistence diagrams/barcodes is discussed, before or alongside the later stability references. This would make the attribution of foundational results on canonical forms/barcodes, their computation, and their stability more accurate.


Second, please add the mentioning of the  recent line of work on comparative/cross-barcodes and topological divergence [2–4]. This should include the NeurIPS 2021 MTD paper, which introduced cross-barcodes/MTop-Divergence and included experiments on time-series data, including market-stock data [2]. It should also include the ICML 2022 RTD paper, which compares two point clouds representing related data, with a point-to-point correspondence and possibly different ambient spaces [3]. Finally, please cite the recent IEEE Access paper, where the cross-barcodes are studied statistically and applied to time-series data [4]. These additions are important because comparative barcode constructions provide an alternative to the standard strategy of computing two separate persistence diagrams and then comparing them by bottleneck or Wasserstein distances, as currently discussed in the manuscript’s background Sections 3.3–3.4.

Third, please implement Reviewer u1ci’s formatting suggestions in the camera-ready version: add hyperlinks to bibliographic references in the main text and taxonomy, add hyperlinks to figures and tables, and fix the remaining dangling pages containing only one or two lines of text.

These requested changes are bibliographic and presentational,  I do not expect new experiments or a major restructuring of the manuscript.

[1] S. A. Barannikov. *The framed Morse complex and its invariants*. Advances in Soviet Mathematics, 21:93–116, 1994.

[2] S. Barannikov, I. Trofimov, G. Sotnikov, E. Trimbach, A. Korotin, A. Filippov, and E. Burnaev. *Manifold Topology Divergence: A Framework for Comparing Data Manifolds*. NeurIPS, 2021.

[3] S. Barannikov, I. Trofimov, N. Balabin, and E. Burnaev. *Representation Topology Divergence: A Method for Comparing Neural Network Representations*. ICML, 2022.

[4] A. Mironenko, E. Burnaev, and S. Barannikov. *The Density of Cross-Persistence Diagrams and Its Applications*. IEEE Access, 14:34320–34338, 2026.

**Audience:**

Yes

**Audience Explanation:**

At least some TMLR readers would be interested in the paper. Persistent homology for time series is a specialized but active topic at the intersection of TDA, machine learning, signal processing, and biomedical/industrial applications. The manuscript reviews a substantial body of work, proposes a unified pipeline-based taxonomy, and can serve as a useful entry point for researchers seeking to understand how persistent homology has been used for time-series analysis.

The likely audience is not broad across all of TMLR, but the paper should be of interest to a meaningful subset of readers working on topological machine learning, representation of temporal data, biomedical signals, dynamical systems, and related applications.

**Claims And Evidence:**

Yes

**Claims Explanation:**

The submission’s main claims are supported by generally convincing and clear evidence. The paper claims to provide a selective review of persistent homology methods for univariate and multivariate time series, covering a substantial body of literature and organizing them by application area, transformation, filtration, and downstream analysis.

The evidence for the survey’s descriptive claims is mostly bibliographic rather than experimental, which is appropriate for a review article.

The mathematical background is also largely appropriate. In particular, the manuscript introduces persistence diagrams and barcodes as equivalent birth–death summaries, and then discusses bottleneck and Wasserstein distances for comparing persistence diagrams/barcodes in Sections 3.3–3.4.

However, the evidence and attribution should be corrected in revision, see below for details. In particular, the omission of the 1994 work on canonical forms, later termed barcodes, including the algorithm for their calculation and their stability result, affects the accurate attribution of several foundational results. Similarly, the omission of the recent line of work on comparative/cross-barcodes and topological divergence leaves out an important alternative to the standard approach of computing two separate persistence diagrams and comparing them by bottleneck or Wasserstein distance, discussed in Sections 3.3–3.4 of the manuscript.

Thus, while I answer Yes because the main survey claims are broadly supported, the manuscript requires minor revision to correct these attributional and bibliographic gaps.